# Influence of partial or total substitution of wheat flour and sunflower oil with Sacha inchi (*Plukenetia volubilis* L.) flour and oil on the quality and nutritional properties of cookies

Fernando E. Alejandro Ruiz[1☯], Julio F. Ortega Jácome[1☯],
Margarita G. Chancay Pinargote[1], José R. Mora[2], Andrea C. Landázuri[3,4,5,6],
Paola Vásconez Duchicela[7], Julio Vásconez Espinoza[7], Pablo Beltrán-Ayala[8],
María J. Andrade-Cuvi[1]*, José M. Álvarez-Suárez[1,9]*

**1** Laboratorio de Investigación en Ingeniería en Alimentos (LabInAli), Departamento de Ingeniería en Alimentos, Colegio de Ciencias e Ingenierías, Universidad San Francisco de Quito USFQ, Quito, Ecuador, **2** Departamento de Ingeniería Química, Colegio de Ciencias e Ingenierías, Universidad San Francisco de Quito USFQ, Quito, Ecuador, **3** Chemical Engineering Department, Applied Circular Engineering & Simulation Group (GICAS), Universidad San Francisco de Quito USFQ, Quito, Ecuador, **4** IBioMed, Universidad San Francisco de Quito USFQ, Quito, Ecuador, **5** Institute for Energy and Materials, Universidad San Francisco de Quito USFQ, Quito, Ecuador, **6** Instituto de Investigaciones Biológicas y Ambientales (Biósfera), Universidad San Francisco de Quito USFQ, Quito, Ecuador, **7** Instituto Superior Universitario Oriente ITSO, Joya de los Sachas, Orellana, Ecuador, **8** School of Economics, Universidad San Francisco de Quito USFQ, Quito, Ecuador, **9** Laboratorio de Bioexploración, Colegio de Ciencias Biológicas y Ambientales, Universidad San Francisco de Quito USFQ, Quito, Ecuador

☯ The authors contributed equally to the present research.
* mjandrade@usfq.edu.ec (MJAC); jalvarez@usfq.edu.ec (JMAS)

## Abstract

### Background

The increasing demand for healthier and more sustainable foods has driven the reformulation of bakery products using alternative ingredients. Sacha inchi (*Plukenetia volubilis* L.), an Amazonian oilseed rich in proteins, unsaturated fatty acids, and bioactive compounds, represents a promising resource to enhance the nutritional and functional quality of baked goods while valorizing agro-industrial by-products.

### Methodology

This study investigated the effects of partially or totally replacing wheat flour and sunflower oil with defatted Sacha inchi oilcake flour and Sacha inchi oil on the nutritional, physicochemical, and functional properties of cookies. Nine formulations were developed, including a control and eight experimental variants. Proximate composition, mineral content, lipid profile, total polyphenols, antioxidant activity (FRAP and DPPH assays), texture, geometry, and color parameters were evaluated.

**Data availability statement:** All relevant data are within the manuscript.

**Funding:** This work was supported by Universidad San Francisco de Quito through an Interdisciplinary Grant (Project ID: 23218) awarded to José M. Alvarez-Suárez. The funders had no role in study design, data collection and analysis, decision to publish, or preparation of the manuscript.

**Competing interests:** The authors have declared that no competing interests exist.

## Results

Cookies containing Sacha inchi flour showed significant increases in protein (up to 19.65%), fat, fiber (6-fold higher), ash, and energy, with a reduction in carbohydrates. Mineral content increased markedly for calcium (10.8-fold), magnesium (10.2-fold), potassium (3-fold), phosphorus (5.4-fold), and zinc (4.6-fold), while iron decreased by 25%. Lipid profiling revealed a higher proportion of polyunsaturated fatty acids, particularly α-linolenic acid (omega-3), and lower saturated fats in cookies containing Sacha inchi oil. The incorporation of Sacha inchi ingredients also increased total polyphenols (up to 1.46-fold) and antioxidant activity (up to 1.99-fold). Texture analysis showed that Sacha inchi flour reduced hardness (up to 5.8 times softer than control), whereas Sacha inchi oil increased firmness (up to 2.4 times). Full replacement of sunflower oil increased cookie diameter and spread ratio, while Sacha inchi flour reduced diameter and increased thickness. Color parameters were also affected, reflecting compositional and Maillard-related changes during baking.

## Conclusions

Replacing wheat flour and sunflower oil with Sacha inchi flour and oil enhanced the nutritional profile, fatty acid composition, and antioxidant capacity of cookies, while modulating their texture and geometry. These findings demonstrate the technological feasibility of using Sacha inchi derivatives as functional ingredients in bakery products, supporting the development of sustainable, health-oriented foods and promoting the valorization of Amazonian crops and by-products.

## Introduction

In recent years, the demand for nutritionally enhanced and functionally innovative food products has grown significantly, driven by increased consumer awareness of healthier diets, environmental sustainability, and the need for greater food system diversification [1]. This trend has motivated the food industry to explore the development of alternative formulations that not only address health-related concerns but also support sustainable production practices. Consumers now increasingly seek foods that offer functional benefits, such as improved digestive health, cardiovascular protection, or enhanced protein content, while also contributing to ethical and environmentally responsible consumption patterns. The baking industry, which has traditionally relied on refined wheat flour and conventional vegetable oils, now faces multiple challenges, including the prevalence of gluten intolerance, rising cases of metabolic disorders [2], and the environmental impact of monoculture-based grain production, such as intensive wheat and maize cultivation, which contributes to soil degradation, biodiversity loss, and increased greenhouse gas emissions [3]. In response, researchers and manufacturers are investigating the substitution of traditional ingredients with alternative raw materials that meet modern nutritional expectations and align with broader goals of environmental sustainability and circular

economy principles [4]. In this context, there is growing interest in the use of non-conventional flours and oils, derived from plant-based sources, agro-industrial by-products, or underutilized crops, as promising candidates for innovation in baked goods.

One promising approach involves the recovery and revalorization of by-products from the food industry to develop new formulations with higher nutritional value and reduced environmental impact. This aligns with the principles of the circular economy, which seeks to minimize waste and promote the efficient use of resources throughout the production chain [5]. By turning food waste and processing residues into valuable functional ingredients, the food industry can improve product sustainability and contribute to closing resource loops. In particular, exotic plant-based ingredients, often rich in bioactive compounds, have gained attention due to their unique nutritional, functional, and sensory properties [6]. Many of these novel ingredients originate from biodiverse ecosystems, such as the Amazon rainforest, and have been traditionally consumed by indigenous communities for their health benefits and nutrient density [7,8]. Their incorporation into modern food systems not only offers sustainable alternatives to conventional ingredients, but also supports the economic valorization of native crops, encourages agro-biodiversity, and helps reduce dependence on resource-intensive monocultures [9]. These strategies contribute to multiple Sustainable Development Goals (SDGs), particularly those related to responsible consumption and production, good health and well-being, and climate action. Among these emerging ingredients, Sacha inchi (*Plukenetia volubilis* L.), a plant native to the Amazon basin and traditionally cultivated by indigenous communities, has attracted significant scientific and commercial interest [10,11]. It is particularly recognized for its remarkable protein and lipid profile, including high levels of omega-3 fatty acids, essential amino acids, and micronutrients [12]. Sacha inchi is recognized for its exceptional chemical composition, with seeds containing 33–54% oil, 24–30% protein, and an optimal ω-6/ω-3 ratio close to 1:1, driven by high levels of α-linolenic acid (~47%) and linoleic acid (~38%) [10]. Recent characterization of Ecuadorian materials confirms this fatty acid profile and additionally shows that the defatted oilcake is highly concentrated in protein (~42%), dietary fiber, and essential minerals, including calcium (2,63 mg/kg), magnesium (3,69 mg/kg), potassium (7,68 mg/kg), and phosphorus (9,14 mg/kg), while retaining meaningful levels of polyphenols and antioxidant activity [13]. These compositional attributes, together with favorable techno-functional properties such as water absorption and swelling capacity, support the use of both Sacha inchi oil and oilcake flour as nutrient-dense, sustainable ingredients for bakery formulations.

Among the various products derived from this plant, one of the most prominent is Sacha inchi oil, which has been widely commercialized for its exceptional fatty acid composition and associated health benefits. However, the defatted seed residue, commonly known as Sacha inchi oil cake, remains an underutilized by-product, despite being rich in protein, dietary fiber, and essential minerals. Nevertheless, despite its promising nutritional composition, the raw oil cake may contain certain antinutritional compounds, including alkaloids, oxalates, nitrates, tannins, thiocyanates, saponins, phytic acid, trypsin inhibitors, and glucosinolates, which can affect protein digestibility and mineral bioavailability if not properly processed. These compounds are, however, largely reduced through technofunctional processing steps such as roasting, drying, extrusion, or baking, as well as hydrothermal treatments (autoclaving or hot air), which have been shown to significantly decrease their levels in Sacha inchi oil cake [14]. After appropriate processing, this by-product offers significant potential for use in food formulations as a partial or total substitute for wheat flour, thereby enhancing the overall nutritional value of baked products [15,16]. Moreover, several studies have reported that both Sacha inchi oil and flour are non-toxic and safe for human consumption, provided that the seed flour undergoes suitable thermal treatment, supporting their potential application in functional food development [17].

Despite the promising nutritional profile of Sacha inchi, several practical challenges limit the commercial utilization of its by-products, including variable extraction efficiencies ranging from 70% to 90% depending on seed origin and processing conditions [18], underdeveloped supply chains, and potential cost disadvantages compared to conventional ingredients. Nevertheless, cookies represent an ideal vehicle for ingredient substitution due to their widespread acceptance across age groups. While interest in Sacha inchi as a functional ingredient is growing, research examining the simultaneous

application of both its flour and oil as direct replacements for traditional baking ingredients remains limited, creating a significant gap in understanding their combined effects in everyday food products. Recent reviews have highlighted that most studies on Sacha inchi focus primarily on the compositional and physicochemical properties of the oil, with comparatively scarce investigations addressing the integration of both fractions—oil and flour—within food matrices or processing systems [19]. This lack of integrated research constrains a comprehensive understanding of the functional and technological synergies between lipid and protein components derived from this underutilized Amazonian crop.

Against this background, our study evaluated how replacing wheat flour and sunflower oil with Sacha inchi derivatives affects cookies' nutritional profile, functional properties, and physical characteristics. We conducted comprehensive bromatological analyses to determine changes in key nutrients (protein, fat, fiber, carbohydrates, and energy content) and examined macro- and micronutrient composition to assess the nutritional impact of these substitutions. Additionally, we analyzed fatty acid profiles to obtain formulations with a health-beneficial lipid composition, characteristic of Sacha inchi oil, and measured polyphenol content and antioxidant capacity to quantify the functional properties these Amazonian ingredients contribute to the final products.

## Materials and methods

### Materials

The raw materials for cookie production were sourced from local supermarkets in Quito, Ecuador. All ingredients were used in their natural state, as commercially available, without prior dehydration or concentration. Wheat flour were used in their dry form, while Sacha inchi oil, sunflower oil, egg, vanilla flavor, sucrose (fine granulated), brownnulated sucrose, honey, sodium bicarbonate, and salt were used as they are commercially sold or available in the marke. The Sacha inchi (*Plukenetia volubilis* L.) defatted cake and oil were donated by members of the Shushufindi community (0.1883° S, 76.6422° W), located in the northern Amazon region of Ecuador, in the province of Sucumbíos. To extract the oil and obtain the defatted oil cake, the peeled Sacha inchi seeds were subjected to oil extraction by cold extrusion at room temperature, which allowed the crude oil to be separated from the defatted solid residue (oil cake). The resulting oil was left to stand for 24 hours in an airtight container, protected from light, at room temperature to promote the settling of suspended particles. After decantation, the clarified upper layer of oil was filtered through a linen cloth and stored in airtight green glass bottles at 4°C. The chemical and bromatological composition of the raw materials employed in this study (defatted Sacha inchi cake flour and Sacha inchi oil) had been previously analyzed and fully characterized by our research group [13]. This prior characterization, conducted under standardized analytical protocols, established the compositional and physicochemical profile of both materials and confirmed their suitability for food formulation purposes. Consequently, both the oil and its defatted by-product were well-defined in terms of their nutritional and chemical attributes, providing a reliable and validated baseline for their use in the present study.

Both the Sacha inchi cake flour and the cookies were ground into a fine powder using a RIRIHONG 2000A multifunction knife mill (Zhejiang, China) and stored at 4 °C in vacuum-sealed plastic bags until further use. The flour, obtained from the milling of the oil cake, was used in the cookie formulations, while the ground cookies were used for bromatological and chemical analyses.

### Formulation of cookies

The cookies were prepared according to the method previously described by Auquiñivin & Castro [20]. A total of nine formulations were developed: one control sample (Ctl), produced using a conventional recipe with wheat flour and sunflower oil, and eight experimental formulations (F1–F8), in which wheat flour and/or sunflower oil were partially or completely substituted with *Sacha inchi* oilcake flour and/or *Sacha inchi* oil. Specifically, F1–F2 included increasing levels of *Sacha inchi* oil (50% and 100%) while maintaining wheat flour unchanged; F3–F5 combined a fixed 50% substitution of wheat flour with oilcake flour with 0%, 50%, and 100% oil, respectively; and F6–F8 consisted of a full (100%) replacement of

wheat flour with oilcake flour, combined with the same incremental gradient of *Sacha inchi* oil. Constant ingredients (egg, sugar, honey, salt, vanilla, and sodium bicarbonate) were maintained across all formulations to ensure that observed differences could be attributed solely to variations in oil and flour composition.

As summarized in **Table 1**, this gradient-based design allowed a systematic evaluation of both individual and combined substitution effects. To enhance visualization and methodological clarity, a schematic flow diagram of the formulation and processing steps is provided in **Fig 1**.

Following the schematic workflow (**Fig 1**), the cookies were prepared as follows: The dry ingredients—wheat flour and/or *Sacha inchi* cake flour, sodium bicarbonate, and salt—were first mixed and kept at room temperature. Brown sugar, white sugar, and shortening and/or *Sacha inchi* oil were then mixed for 2 minutes using a Classic™ Max Watts 275 mixer (KitchenAid®, Michigan, USA). Vanilla extract and honey were subsequently incorporated and mixed for an additional 2 minutes. The previously blended dry ingredients were then added to this mixture, and homogenization continued until a uniform dough was obtained. The dough was refrigerated at 4 ºC for 10 minutes and then portioned into 5 g units, which were placed on a pre-greased oblong tray. Each portion was rolled once back and forth to reach a thickness of 1 cm and shaped using 4 cm diameter molds. The cookies were baked at 180 °C for 12 minutes in a commercial convection oven (Prática Technipan, Brazil). After baking, the cookies were allowed to cool at room temperature for 30 minutes, wrapped in aluminum foil, and stored at room temperature for 24 h prior to analysis.

## Proximate analysis

Proximate composition analyses of pre-ground cookies were conducted following the standardized protocols outlined in the AOAC Official Methods of Analysis. Color was evaluated using a Konica Minolta CR-400 colorimeter (Konica Minolta Sensing Americas, Inc., NY, EEUU) in the CIELAB color space [21]. Moisture content (%) was determined through

**Table 1. Composition of control and Sacha inchi–based cookie formulations.**

| Formulation group | Formulation code | Variable ingredients (% w/w) | Fixed ingredients (% w/w) |
|---|---|---|---|
| Control formulation | Ctl | Wheat flour (100), sunflower oil (66.67) | Egg (27.78), sucrose (27.78), brown sugar (55.56), honey (2.33), vanilla (1.00), sodium bicarbonate (1.39), salt (1.39) |
| Partial replacement of sunflower oil with Sacha inchi oil (flour unchanged) | F1 | Wheat flour (100), Sacha inchi oil (33.33), sunflower oil (33.33) | |
| | F2 | Wheat flour (100), Sacha inchi oil (66.67), sunflower oil (0) | |
| 50% replacement of wheat flour with Sacha inchi oilcake flour + variable oil substitution | F3 | Wheat flour (50), Sacha inchi oilcake flour (50), sunflower oil (66.67) | |
| | F4 | Wheat flour (50), Sacha inchi oilcake flour (50), Sacha inchi oil (33.33), sunflower oil (33.33) | |
| | F5 | Wheat flour (50), Sacha inchi oilcake flour (50), Sacha inchi oil (66.67), sunflower oil (0) | |
| 100% replacement of wheat flour with Sacha inchi oilcake flour + variable oil substitution | F6 | Sacha inchi oilcake flour (100), sunflower oil (66.67) | |
| | F7 | Sacha inchi oilcake flour (100), Sacha inchi oil (33.33), sunflower oil (33.33) | |
| | F8 | Sacha inchi oilcake flour (100), Sacha inchi oil (66.67), sunflower oil (0) | |

Fixed ingredients (egg, sucrose, brown sugar, honey, vanilla, sodium bicarbonate, and salt) were maintained constant across all formulations to isolate the effects of replacing wheat flour and/or sunflower oil with Sacha inchi oil and oilcake flour.

Ctl = Control formulation without Sacha inchi ingredients; F1–F2 = Partial substitution of sunflower oil with Sacha inchi oil; F3–F5 = 50% replacement of wheat flour with Sacha inchi oilcake flour combined with increasing levels of Sacha inchi oil; F6–F8 = 100% replacement of wheat flour with Sacha inchi oilcake flour combined with increasing levels of Sacha inchi oil.

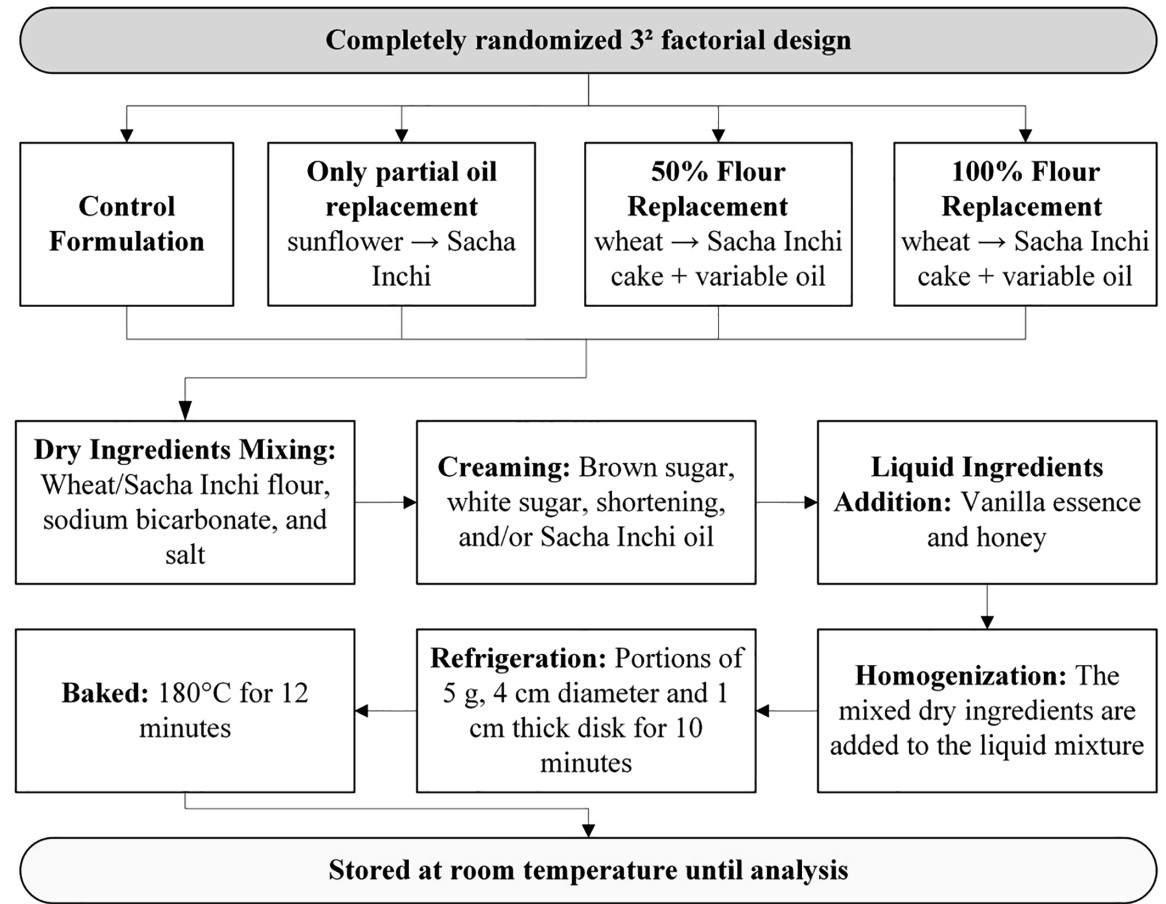

**Fig 1. Flowchart illustrating the formulation and processing workflow of cookies enriched with Sacha inchi oil and/or defatted Sacha inchi cake flour, including ingredient preparation, mixing, dough formation, molding, and baking.**

gravimetric analysis (AOAC 925.10) [22], whereas water activity ($a_w$) was measured using a HygroLab C1 water activity meter (Rotronic, Grindelstrasse, CH-Switzerland). Ash content (%) was quantified by incinerating 3 g of samples in a muffle furnace at 550 ± 1 °C overnight (AOAC 923.03) [23]. Additionally, total crude fiber content (%) was assessed through sequential extraction with acidic and alkaline solutions (AOAC 978.10) [24] whyle the fraction corresponding to soluble (%) and insoluble fiber (%) using methods proposed by the Cereals & Grains Association (AACC 32–05.01) [25] and AOAC 985:29 [26]. Meanwhile, lipid content (%) was determined via Soxhlet extraction using petroleum ether as the solvent (AOAC 920:39) [27], whereas protein content (%) was analyzed using the Kjeldahl method, applying a nitrogen-to-protein conversion factor of 6.25 (AOAC 2001.11) [28]. Finally, carbohydrate content (g/100 g) and energy value (kcal/100 g) were calculated using Equations (1) and (2), as described by Plustea et al. [29], respectively, as shown below:

$$\text{Carbohydrate (g/100 g)} = 100 - (\text{fat} + \text{protein} + \text{water} + \text{ash}) \tag{1}$$

$$\text{Energy value (kcal/100 g)} = (\text{fat} \times 9) + (\text{carbohydrate} \times 4) + (\text{protein} \times 4) \tag{2}$$

Conversion factors of 9 kcal/g for fat and 4 kcal/g for protein and carbohydrates were applied according to the Atwater system [30].

## Gas chromatography-mass spectrometry (GC-MS) analysis of fatty acid composition

Fatty acid content analysis was performed following the internal standard method described by Ponce et al. and based on AC Analytical Controls Application Note 1410 [31,32]. For this procedure, a transesterification reaction was first carried out on $0.5 \pm 0.01$ g of the oil extracted from the cookies, obtained by Soxhlet extraction with petroleum ether, using methanol (methanol: fat molar ratio: 14:1) and 1% m/m potassium hydroxide (KOH) as catalyst. To facilitate quantification, methyl dodecanoate (2.0 mg/mL) was used as an internal standard. The resulting fatty acid methyl esters (FAMEs) were dissolved in hexane before analysis.

For GC-MS analysis, a Shimadzu GCMS-QP-2010 Ultra Plus equipped with an AOC-20i/s autosampler and an auto-injector was used. The separation was performed on a DB-WAX gas chromatography (GC) capillary column (column thickness: 0.25 µm, length: 30.0 m, internal diameter: 0.25 µm). The oven configuration was set as follows: an initial temperature of 120 °C with a hold time of 2 min, followed by a ramp to 190 °C at 15 °C/min and a hold time of 4 min, then a ramp to 240 °C at 5 °C/min and a hold time of 10 min, and finally a ramp to 250 °C at 5 °C/min and a hold time of 3 min. The injector temperature was set at 250 °C, and 1 µL of the sample was injected with a 1:4 split mode. Helium was used as the carrier gas, considering the linear velocity for a column flow of 0.98. Mass detection was configured to start at an $m/z$ of 35.00 and end at an $m/z$ of 700.00. Calibration curves were generated using individual standard solutions (1.45–2.55 mg/L) of palmitic, oleic, and linoleic acids. The corresponding regression equations and analytical performance parameters were as follows: palmitic acid, $y = (0.9712 \pm 0.0119) x + (0.0377 \pm 0.0027)$, $R^2 = 0.9930$; oleic acid, $y = (0.9951 \pm 0.0086) x + (0.0245 \pm 0.0017)$, $R^2 = 0.9994$; and linoleic acid, $y = (0.9315 \pm 0.0144) x + (0.0442 \pm 0.0015)$, $R^2 = 0.9982$. The mass ratio is "x", and the area ratio is "y". Based on the response factor of 1:1 for the previous three fatty acids (slope of about 1), which implies that $mi/mt = Ai/At$ (mass ratio is equal to area ratio), it was possible to quantify the other fatty acids as:

$$\% \ Fai \ = \ (Ai/At) \ x \ 100$$

(3)

where % Fai is the percentage of the corresponding fatty acid "i", Ai is the area of the given fatty acid, and At is the total area ($At = \Sigma Ai$). In the calibration curve equations, the mass ratio is "x", and the area ratio is "y". The results were expressed in grams of fatty acids per 100 g of oil (g/100 g for the oil).

## Determination of macro- and microelements using inductively coupled plasma optical emission spectrometry (ICP-OES)

Briefly, $0.5 \pm 0.1$ g of each sample was placed in a digestion vessel, followed by the addition of 9 mL of nitric acid (TraceMetal Grade™, Fisher Chemical™) and 0.4 mL of hydrogen peroxide (30%, Fisher Chemical™). The mixture was subjected to microwave-assisted digestion, and the resulting solution was filtered through a 0.45 µm PTFE membrane and diluted to a final volume of 50 mL with deionized water. For quality control, a spiked sample was prepared using the same procedure, yielding recoveries ranging from 84.70% to 99.96%. The determination of macro- and microelements was performed using an ICP-OES system (iCAP™ 7400 Duo ICP-OES spectrometer, Thermo Scientific™, Germany), operated with Thermo Scientific™ Qtegra™ Intelligent Scientific Data Solution™ (ISDS) software, as previously applied in the characterization of Sacha inchi flour [13]. Calibration curves were established using an ICP VIII multi-element standard solution. The analytical calibration equations for each element were as follows: Al: $y = 1.445x + 0.189$ ($R^2 = 0.948$); As: $y = 8.579x - 0.068$ ($R^2 = 0.997$); B: $y = 160.426x + 22.344$ ($R^2 = 0.999$); Ba: $y = 19.815x + 112.551$ ($R^2 = 1.000$); Ca: $y = 16.213x + 0.807$ ($R^2 = 0.985$); Cd: $y = 500.056x + 0.321$ ($R^2 = 0.999$); Cr: $y = 256.663x - 3.750$ ($R^2 = 1.000$); Cu: $y = 245.835x - 4.001$ ($R^2 = 1.000$); Fe: $y = 203.326x - 0.004$ ($R^2 = 0.991$); K: $y = 261.916x - 20.859$ ($R^2 = 0.997$); Mg: $y = 9.637x + 7.296$ ($R^2 = 0.999$); Mn: $y = 1.776x - 0.167$ ($R^2 = 1.000$); Na: $y = 940.978x - 41.059$ ($R^2 = 0.984$); Ni: $y = 48.114x + 7.620$ ($R^2 = 0.998$); P: $y = 3.844x + 0.230$ ($R^2 = 0.989$); Pb: $y = 24.809x - 0.146$ ($R^2 = 0.998$); S: $y = 4.400x + 0.369$ ($R^2 = 0.991$);

Se: y = 5.617x + 0.906 (R² = 0.995); Si: y = 94.944x + 2.923 (R² = 0.999); Sr: y = 31.642x − 6.325 (R² = 1.000); V: y = 637.164x − 15.884 (R² = 1.000); Zn: y = 282.680x + 3.641 (R² = 0.987). All results were expressed as mg/100 g dry weight (DW).

## Determination of polyphenols and total antioxidant activity

To evaluate polyphenol content and total antioxidant activity, a hydroalcoholic extraction was conducted based on the methodology previously reported by Sokmen et al. [33]. In summary, 0.5 ± 0.01 g of ground cookies, previously defatted, was extracted with 5 mL of an HClcon/methanol/water mixture (1/80:19, *v/v*). The solution was stirred in the dark at room temperature for 2 hours using a magnetic stirrer. Afterward, the sample was centrifuged (HERMLE Z 206 A, Wehingen, DE) at 20°C for 10 minutes at 10,000 rpm, filtered through a Whatman˚ Grade 1 qualitative filter paper (Sigma Aldrich, St. Louis, MO, USA), and stored at −20 °C until further analysis.

The determination of total polyphenol content (TPC) was carried out spectrophotometrically using an i3 UV-VIS Spectrophotometer (Hanon Instruments, Shandong, China) at 760 nm, following the Folin-Ciocalteu method [34]. A standard curve was generated using gallic acid (30,6–102,1 mg/L), yielding the regression equation y = 1,6509x + 0,007 (R² = 0,999). Results were expressed as milligrams of gallic acid equivalents per 100 grams of fresh weight (mg GAE/100 g FW) of cookies.

On the other hand, total antioxidant capacity was simultaneously evaluated using two spectrophotometric methods: the Ferric Reducing Antioxidant Power (FRAP) assay at 593 nm [35] and the 2,2-Diphenyl-1-picrylhydrazyl (DPPH) assay at 517 nm [36]. In both assays, Trolox was used as standard to establish the calibration curve (for FRAP: 12,5–300 µM, y = 0,0028x + 0,023, R² = 0,9938; for DPPH: 0,1–2 mM, y = −3,3164x + 2,2931, R² = 0,972), and the results were expressed as millimoles of Trolox equivalents (mM TEq) per gram of fresh weight (mM TEq/g FW) of cookie.

**Physical analysis of cookies.** Diameter (D), Thickness (T) and Spread ratio

Six cookies from each treatment group were measured using a vernier caliper with a precision of ± 0.02 mm, following the standard procedures outlined by the American Association of Cereal Chemists (AACC) [37]. The diameter (D) and thickness (T) of each cookie were recorded, and the mean values were calculated and expressed in centimeters (cm). To determine the spreading ratio, the average diameter was divided by the average thickness (D/T), as previously described by Korese et al. [38].

**Color determination.** The surface color of the cookies was measured using a colorimeter (CR-400, Konica Minolta Inc., Japan), which was previously calibrated with a standard white calibration plate under D65 illuminant conditions (Y = 80.1, x = 0.3219, y = 0.3394). Color measurements were recorded using the CIE L*ab* color space, where $L*$ represents lightness, $a*$ indicates the red-green axis, and $b*$ represents the yellow-blue axis [21]. For each treatment, a minimum of six cookie samples were analyzed to ensure measurement accuracy and reproducibility. The total color difference (ΔE) compared to the control formulation was calculated using the following equation:

$$\Delta L = \sqrt{\left(L_0^* - L^*\right)^2 + \left(a_0^* - a^*\right)^2 + \left(b_0^* - b^*\right)^2} \tag{4}$$

where $L_0^*$, $a_0^*$ and $b_0^*$ are the color values of the control sample, and $L*$, $a*$, $b*$ correspond to the color values of the treated cookies. This parameter provides an objective quantification of perceptible color differences resulting from ingredient or processing variations.

**Hardness.** The texture analysis of cookies was determined by using texture analyzer (TX-700-UK02/2023, LAMY Rheology Instruments, France). As described by Baltakesmez and Mısır [39], hardness of the cookies was evaluated by using a cylindrical probe (TX-CY2H355) at a speed of 2.5 mm/s and compression distance was set at 4 mm. The triggering force was 50 g using a load cell of 250 N. Data were recorded in a RheoTex Software and the results were expressed in Newtons (N).

## Design of experiment and statistical analysis

A completely randomized 3² factorial design was used to evaluate the effect of two factors: flour and sunflower oil, which were replaced with Sacha inchi oil cake and Sacha inchi oil, respectively. Each factor was tested at three levels (0%, 50%, and 100%). The design included two factors, three levels, and three replicates, resulting in 27 randomized experimental units. The experimental design and randomization were conducted using Minitab 19. The samples were analyzed in triplicate, and results were reported as mean ± standard deviation (SD). Data were analyzed by analysis of variance (ANOVA), and means were compared using Tukey's test at a significance level of 0.05, using INFOSTAT statistical software (Facultad de Ciencias Agropecuarias, Universidad Nacional de Córdoba, Argentina). The samples were analyzed in triplicate, and results were reported as mean ± standard deviation (SD).

## Results and discussion

### Bromatological analysis of cookies formulated with Sacha inchi flour and oil

The analysis of the proximal and chemical composition of cookies formulated by replacing wheat flour and sunflower oil with defatted Sacha inchi oil cake flour and Sacha inchi oil showed significant changes in various compositional and functional parameters (Table 2). According to the results of the bromatological analysis, the moisture content increased significantly ($p ≤ 0.05$) in formulations with higher percentages of Sacha inchi oil cake flour and oil, reaching an increase of up to 1.16 times in the formulation with 100% flour and oil replacement compared to the control cookie. However, no significant differences were observed among formulations with the same level of flour substitution but different levels of oil substitution. Given that Sacha inchi oil cake contains a residual oil content of up to 25%, the results suggest that an increase in the replacement percentage of wheat flour with Sacha inchi oil cake flour effectively influences the fat content of the cookies and, consequently, their ability to retain moisture. Regarding the water activity values, the results revealed a significant decrease ($p ≤ 0.05$) as the replacement percentage of wheat flour and sunflower oil increased, compared to formulations with a lower degree of substitution. Similarly, to moisture content, no significant differences were observed among formulations with the same level of flour substitution but different levels of oil substitution. These results suggest that a higher substitution of wheat flour and sunflower oil with Sacha inchi oilcake flour and Sacha inchi oil is suitable for preventing microbial growth and extending product stability. The behavior observed in both moisture content and water activity can be attributed to several physicochemical factors. A formulation with a higher fat content can retain more water due to physicochemical and structural mechanisms. Fat acts as a structuring agent, modifying water distribution within the food matrix and reducing its loss during baking and storage [40]. Furthermore, its hydrophobic nature allows the formation of a barrier that limits water evaporation, effectively retaining moisture within the product's structure [41]. Additionally, fat can interfere with protein network formation and coat starch granules, affecting water absorption and retention. Its presence lowers water activity by reducing free water mobility, thereby contributing to decreased evaporation. Finally, in high-fat products, the texture is generally softer and less porous, which makes water retention more effective compared to low-fat cookies, which tend to be drier and more porous, thus promoting moisture loss [40]. In line with the results described above, the ash content increased significantly with flour substitution, reaching a maximum of 2.38% in the formulation with 100% substitution of wheat flour by Sacha inchi cake flour, with an increase of 2.27-fold with respect to the control sample (1.05%). These results are consistent with previous findings, where the incorporation of sacha inchi cake flour in the preparation of Peruvian traditional flatbread resulted in a significant increase in ash content in the sacha inchi flour-based formulations [16]. Similarly, the protein content showed a significant increase ($p ≤ 0.05$) when wheat flour was substituted by Sacha inchi oil cake flour. The formulations with 100% flour substitution presented values of 19.34–19.65%, more than 2.2 times compared to 8.83% in the control sample. This increase is attributable to the high protein content of the defatted sacha inchi cake flour, in line with the results of the previous bromatological analysis of this flour [13], as well as with findings reported in the literatura [10,42]. This makes these cookies a potentially relevant source of plant-based

**Table 2. Proximal composition and chemical composition of control and cookies with Sacha inchi oil cake flour and oil substitutes.**

| Analysis | Ctl | F1 | F2 | F3 | F4 | F5 | F6 | F7 | F8 |
|---|---|---|---|---|---|---|---|---|---|
| **Cookie's formulations** | | | | | | | | | |
| ***Proximate*** | | | | | | | | | |
| Moisture (%) | 2.22±0.01[a] | 2.21±0.03[a] | 2.22±0.03[a] | 2.38±0.01[b] | 2.35±0.03[b] | 2.36±0.03[b] | 2.54±0.03[c] | 2.55±0.04[c] | 2.57±0.07[c] |
| Water activity ($a_w$) | 0.52±0.03[a] | 0.54±0.02[a] | 0.52±0.05[a] | 0.39±0.02[b] | 0.41±0.01[b] | 0.42±0.03[b] | 0.26±0.01[c] | 0.25±0.02[c] | 0.25±0.01[c] |
| Ash (%) | 1.05±0.05[a] | 1.06±0.05[a] | 1.07±0.02[a] | 1.61±0.01[b] | 1.65±0.01[b] | 1.70±0.02[b] | 2.39±0.02[c] | 2.34±0.05[c] | 2.38±0.07[c] |
| Protein (%) | 8.83±0.22[a] | 8.09±0.72[a] | 7.47±1.23[a] | 15.18±0.31[b] | 14.54±0.25[b] | 14.19±0.39[b] | 19.45±0.57[c] | 19.49±1.14[c] | 19.34±0.50[c] |
| Fat (%) | 19.18±0.68[a] | 22.86±0.44[a] | 22.77±1.82[a] | 27.70±0.93[b] | 28.89±1.09[b] | 28.70±0.46[b] | 31.60±0.87[c] | 34.68±0.36[c] | 32.77±1.82[c] |
| Crude fiber (%) | 0.55±0.02[a] | 0.55±0.05[a] | 0.57±0.04[a] | 1.81±0.12[b] | 1.84±0.05[b] | 1.80±0.08[b] | 3.22±0.17[c] | 3.29±0.11[c] | 3.25±0.10[c] |
| Total carbohydrate (%) | 68.72±0.45[a] | 65.77±0.32[a] | 66.47±1.90[a] | 54.19±1.23[b] | 52.56±1.02[b] | 53.05±0.69[b] | 43.93±0.43[c] | 40.93±0.85[c] | 42.93±1.81[c] |
| Energy value (kcal/100g) | 482.82±3.54[a] | 501.10±2.24[b] | 500.71±6.43[b] | 517.54±4.66[c] | 528.45±5.34[d] | 527.28±2.31[d] | 538.72±4.35[d] | 553.84±1.74[e] | 544.06±9.62[d] |
| Dietary fiber (%) | | | | | | | | | |
| Soluble fiber (%) | 1.34±0.25[a] | 1.55±0.12[a] | 1.35±0.23[a] | 0.95±0.13[ab] | 1.40±0.23[a] | 2.52±0.80[b] | 1.67±0.23[ab] | 0.75±0.10[a] | 2.02±0.28[a] |
| Insoluble fiber (%) | 1.02±0.04[a] | 1.03±0.43[a] | 1.05±0.04[a] | 6.36±0.34[b] | 8.31±0.45[b] | 7.76±0.32[b] | 12.98±0.60[c] | 22.25±1.41[d] | 71.59±0.87[e] |
| ***Fatty acids composition (g/100g)*** | | | | | | | | | |
| Palmitic acid (C16) | 2.10±0.61[a] | 1.86±0.79[b] | 1.91±0.90[b] | 0.92±0.45[c] | 1.88±0.82[b] | 1.62±0.10[b] | 1.73±0.48[b] | 1.69±0.46[b] | 1.72±0.65[b] |
| Stearic acid (C18) | 1.28±0.34[a] | 1.23±0.53[a] | 1.27±0.16[a] | 0.57±0.02[b] | 1.30±0.15[a] | 2.41±0.12[c] | 1.84±0.11[d] | 1.12±0.19[e] | 1.13±0.23[e] |
| Oleic acid (C18) | 12.78±2.33[a] | 8.33±1.82[b] | 5.87±0.57[c] | 6.76±0.96[d] | 6.12±0.52[d] | 6.44±0.85[d] | 5.83±0.69[a] | 5.56±0.23[a] | 3.43±0.86[c] |
| Linoleic acid (C18) | 12.93±1.87[a] | 10.46±1.78[b] | 12.51±1.68[a] | 9.89±0.53[c] | 10.14±1.58[c] | 10.90±1.86[a] | 9.74±1.07[c] | 9.75±0.40[c] | 11.53±1.26[b] |
| Linolenic acid (C18) | 2.55±0.11[a] | 4.89±0.23[b] | 7.23±1.13[c] | 2.59±0.24[a] | 6.23±0.77[c] | 9.34±1.17[d] | 4.32±0.99[b] | 6.63±0.02[c] | 7.88±1.91[c] |
| ***Macro and microelements (mg/100g)*** | | | | | | | | | |
| *Macroelements* | | | | | | | | | |
| Calcium (Ca) | 15.19±0.16[a] | 14.86±0.76[a] | 15.11±0.50[a] | 80.76±1.54[b] | 76.84±4.21[b] | 78.01±1.23[b] | 157.13±9.08[c] | 159.36±5.18[c] | 164.22±3.20[c] |
| Magnesium (Mg) | 15.47±0.48[a] | 14.30±0.25[a] | 14.90±0.19[a] | 75.63±1.49[b] | 73.03±1.01[b] | 75.40±1.73[b] | 152.19±9.03[c] | 154.23±3.62[c] | 157.26±3.83[c] |
| Potassium (K) | 83.59±3.64[a] | 81.61±1.79[a] | 80.16±2.58[a] | 180.43±3.24[b] | 177.13±5.64[b] | 179.72±4.09[b] | 234.21±4.39[c] | 236.68±8.13[c] | 240.98±12.67[c] |
| Sodium (Na) | 308.63±2.37[a] | 312.55±9.87[a] | 310.43±4.36[a] | 308.74±14.46[a] | 305.93±5.73[a] | 311.39±7.19[a] | 306.61±12.59[a] | 307.19±5.51[a] | 313.62±9.94[a] |
| Phosphorus (P) | 75.24±3.03[a] | 72.53±2.88[a] | 73.57±3.64[a] | 238.21±3.94[b] | 246.48±3.41[b] | 239.54±4.27[b] | 399.59±6.87[b] | 401.10±11.13[b] | 404.36±14.45[b] |
| *Microelements* | | | | | | | | | |
| Iron (Fe) | 2.10±0.05[a] | 2.14±0.01[a] | 2.11±0.22[a] | 1.58±0.09[b] | 1.64±0.14[b] | 1.65±0.03[b] | 1.55±0.09[b] | 1.49±0.07[b] | 1.56±0.12[b] |
| Zinc (Zn) | 0.57±0.02[a] | 0.52±0.01[a] | 0.50±0.01[a] | 1.76±0.02[b] | 1.66±0.02[b] | 1.69±0.01[b] | 2.64±0.04[c] | 2.57±0.11[c] | 2.61±0.44[c] |
| ***Bioactivity*** | | | | | | | | | |
| TPC (mg GAEq/100 g) | 4.79±0.13[a] | 4.90±0.12[a] | 5.05±0.10[a] | 6.53±0.35[b] | 6.85±0.03[b] | 6.90±0.07[b] | 6.76±0.06[c] | 6.89±0.05[c] | 6.99±0.07[c] |
| FRAP assay (mM TEq/g FW) | 51.05±1.29[a] | 71.55±3.5[b] | 85.04±4.15[b] | 55.45±1.92[c] | 66.19±1.41[d] | 78.57±2.55[e] | 70.89±2.78[f] | 78.55±2.78[f] | 101.55±5.40[g] |
| DPPH assay (mM TEq/g FW) | 21.12±0.99[a] | 24.32±1.50[b] | 23.42±1.53[b] | 18.67±0.70[c] | 20.23±1.38[c] | 20.04±1.14[a] | 16.99±1.20[d] | 17.57±1.19[d] | 17.17±2.30[d] |

Data represent the mean±standard deviation. Different superscript letters indicate significantly different mean values ($p < 0.05$).

**Ctl:** 0% sacha inchi oil cake flour, 0% sacha inchi oil; **F1:** 0% oil cake flour, 50% sacha inchi oil; **F2:** 0% oil cake flour, 100% sacha inchi oil; **F3:** 50% oil cake flour, 0% sacha inchi oil; **F4:** 50% oil cake flour, 50% sacha inchi oil; **F5:** 50% oil cake flour, 100% sacha inchi oil; **F6:** 100% oil cake flour, 0% sacha inchi oil; **F7:** 100% oil cake flour, 50% sacha inchi oil; **F8:** 100% oil cake flour, 100% sacha inchi oil.

protein. These findings are also consistent with previous reports on the effects of sacha inchi flour in bread substitutes [16]. Similar to the case of previous parameters within the groups with the same flour substitution, no statistically significant differences were observed between the different levels of oil substitution. This suggests that the Sacha inchi defatted cake flour is the main determining factor in the protein content. On the other hand, the fat content increased significantly with the substitution of wheat flour by Sacha inchi cake flour, representing a 1.81-fold increase in the formulation with 100% substitution, consistent with previous reports where the incorporation of sacha inchi flour similarly increased the fat content in the formulations [16]. In the treatments with the same percentage of flour substitution, an increase proportional to the level of oil substitution was observed, suggesting that the balance of Sacha inchi oil and defatted cake flour influences fat retention in the formulation. Another compositional result to highlight is the carbohydrate content in each of the formulations. In this case, carbohydrate content decreased with flour substitution, showing a 1.67-fold reduction, while the energy value increased in all formulations with flour and oil substitution, reaching a 1.15-fold increase in the 100% wheat flour substitution. These results are in line with previous findings on the use of sacha inchi flour in bakery formulations, where similar trends in carbohydrate reduction and energy increase were observed [16]. This result, in turn, can be explained by the higher fat content in the formulations with sacha inchi oil cake flour substitution, as previously observed in the bromatological analysis of this flour [13], which contains a residual fat content of up to 25%. From the point of view of fiber contribution by the different formulations, an improvement in the content of this component was observed. Specifically, the total crude fiber content increased by up to 5.98 times compared to the control when 100% defatted sacha inchi oilcake flour was incorporated, consistent with previous reports on the use of defatted sacha inchi cake flour in baked products [16]. Insoluble fiber showed a significant increase, up to 70.2 times, which highlights the high fiber content in the defatted Sacha inchi oilcake flour. Significant differences were observed between the treatments with different flour substitution. Soluble fiber showed a more variable behavior, reaching up to 1.88 times more in formulations with the highest level of flour substitution, compared to the control. From a functional and nutritional perspective, the content and contribution of dietary fiber in foods is of great importance due to its beneficial effect on digestive and metabolic health. Insoluble fiber, which showed the greatest increase in these formulations, favors intestinal transit, prevents constipation and contributes to the health of the intestinal microbiome [43,44]. On the other hand, soluble fiber has the ability to retain water and form gels, which can positively influence the feeling of satiety and the regulation of glucose and cholesterol in the blood [45]. In addition, adequate consumption of dietary fiber has been associated with a reduced risk of chronic diseases such as type 2 diabetes, cardiovascular diseases and some types of cancer [46]. Therefore, the increase in fiber content in these formulations not only improves their nutritional value but also enhances their functionality as a food with potential health benefits.

### Fatty acid profile and content of cookies formulated with Sacha inchi flour and oil

The fatty acid composition of the formulated cookies varied significantly ($p \leq 0.05$) depending on the level of substitution with sacha inchi oil and oil cake flour (**Table 2**), consistent with previous findings [16]. The results indicate that the incorporation of Sacha inchi oil and oil cake flour influenced the profiles of saturated fatty acids (SFA), monounsaturated fatty acids (MUFA), and polyunsaturated fatty acids (PUFA) in the final formulation.

The content of saturated fatty acids, mainly palmitic acid (C16:0) and stearic acid (C18:0), showed variations among formulations. Formulations with a higher proportion of Sacha inchi oil generally had lower levels of palmitic acid than the control formulation (without replacement, F1) ($p \leq 0.05$). This is consistent with the lower SFA content of Sacha inchi oil compared to sunflower oil [13], which typically contains higher amounts of these fatty acids [47] The inclusion of oil cake flour did not significantly increase SFA content, suggesting that it retains only minimal amounts of these fatty acids. The stearic acid content also showed significant ($p \leq 0.05$) variations. Formulations with higher levels of Sacha inchi oil cake flour exhibited a significant reduction ($p \leq 0.05$) in this fatty acid compared to the control, except for formulation with 50% substitution of flour and 100% sunflower oil substitution, which showed the highest stearic acid content, possibly due to

specific interactions during formulation. The oleic acid (C18:1) content varied significantly among formulations ($p \leq 0.05$) and was primarily influenced by the level of substitution with Sacha inchi oil. The control formulation, 0% substitution, exhibited the highest oleic acid content, which aligns with the natural composition of sunflower oil, known for its high oleic acid content [47]. In contrast, increased Sacha inchi oil substitution led to a significant ($p \leq 0.05$) reduction in oleic acid levels, with the lowest content observed in formulations with the highest sunflower oil substitution. This is expected since Sacha inchi oil contains lower proportions of this monounsaturated fatty acid compared to sunflower oil [13]. On the other hand, the incorporation of Sacha inchi cake flour alone did not significantly alter the oleic acid content, reinforcing the idea that the cake fraction retains only limited amounts of this fatty acid. Regarding polyunsaturated fatty acids (PUFAs), substitution with Sacha inchi oil significantly increased ($p \leq 0.05$) the linolenic acid (C18:3) content. The highest levels were observed in formulations with 100% Sacha inchi oil replacement. This increase is consistent with the composition of Sacha inchi oil (46.92% linolenic acid) (**Table 2**), highlighting its potential as an excellent source of omega-3 fatty acids for functional food applications. In contrast, formulations with no sunflower oil substitution had elevated levels of linoleic acid (C18:2), which is abundant in sunflower oil [47]. Finally, formulations with higher levels of Sacha inchi oil showed a slight but significant ($p \leq 0.05$) reduction in linoleic acid. The inclusion of oilcake did not significantly contribute to the increase in linoleic acid levels, confirming that sunflower oil is the main source of this PUFA in the formulations.

The changes in fatty acid composition due to the inclusion of Sacha inchi oil and oilcake have significant nutritional implications. The increased omega-3 (linolenic acid) content in formulations with Sacha inchi oil is beneficial for cardiovascular health, as omega-3 fatty acids are known for their anti-inflammatory properties. They can help lower triglycerides, reduce the risk of arrhythmias, and slow plaque buildup. Additionally, omega-3 fatty acids have been shown to reduce cardiovascular mortality and improve cardiovascular outcomes [48]. Thus, the reduction in SFA and the increase in PUFA, particularly in formulations with Sacha inchi oil substitution, suggest that these formulations may provide a healthier lipid profile compared to traditional formulations based solely on sunflower oil and wheat flour. No long-chain omega-3 fatty acids (such as EPA and DHA) or long-chain omega-6 fatty acids (e.g., arachidonic acid) were detected in any of the formulations. This is consistent with the previously reported fatty acid profile of Sacha inchi oil, which is predominantly α-linolenic acid (C18:3, ω-3) together with linoleic acid (C18:2, ω-6), and lacks long-chain omega-3 PUFA such as EPA and DHA. Similarly, no short-chain fatty acids (C4:0–C10:0) were observed in our chromatograms, which is also consistent with previous reports for oils from this seed [13,10,49,50].

## Macro- and microelement content of cookies formulated with Sacha inchi flour and oil

The analysis of macro- and microelement contents in cookie formulations with different levels of wheat flour replacement by Sacha inchi flour and sunflower oil replacement by Sacha inchi oil reveals significant patterns in the mineral composition of the final product (**Table 2**). Within the macroelements, the levels of calcium (Ca), magnesium (Mg), potassium (K), and phosphorus (P) increased significantly ($p \leq 0.05$) with the incorporation of Sacha inchi flour in the formulation. These results are consistent with the high content reported of theses macroelements in the Sacha inchi cake flour [13] and previous finding [10,42], which explains its increase as the proportion of Sacha inchi cake flour in the mix increases (**Table 2**). In fact, in formulations without substitution, the Ca and Mg contents remained low (~15 mg/100 g), whereas in formulations with 50% Sacha inchi flour substitution, these values quintupled, reaching approximately 76–80 mg/100 g. This corresponds to an increase of 5.3-fold for calcium and 5-fold for magnesium compared to the control formulations (without substitution). In formulations with 100% substitution, the values exceeded 150 mg/100 g, representing an increase of 10.8-fold for calcium and 10.2-fold for magnesium relative to the non-substituted formulations. Potassium (K) followed a similar trend, with a progressive increase from approximately 80 mg/100 g in the formulations without substitution to values exceeding 180 mg/100 g in those with 50% substitution, representing an increase of 2.25-fold. In formulations with 100% substitution, potassium reached up to 240 mg/100 g, representing a 3-fold increase compared to formulations without Sacha inchi flour. Phosphorus (P) levels also exhibited a substantial increase with the replacement of wheat flour by

Sacha inchi flour. In the formulations with the highest Sacha inchi flour content, phosphorus reached up to 404 mg/100 g (F9), representing a 5.4-fold increase compared to the non-substituted formulations. These results are consistent with the high content reported in the sacha inchi cake flour [13] and previous finding [10,42], suggesting that sacha inchi flour is a rich source of this essential mineral. In contrast, sodium (Na) levels remained stable across all formulations, with values close to 310 mg/100 g, indicating that the substitution of both flour and oil did not significantly affect the sodium content in the final product.

Regarding microelements, iron (Fe), unlike macrominerals, exhibited a significant decreasing ($p \leq 0.05$) trend with the progressive replacement of wheat flour by Sacha inchi oil cake flour. In formulations without substitution, iron content exceeded 2 mg/100 g, whereas in formulations with 50% and 100% substitution, the values declined to approximately 1.5 mg/100 g. This corresponds to a 25% reduction compared to the non-substituted formulation, suggesting that wheat flour was a richer source of iron than Sacha inchi flour. Zinc (Zn), on the other hand, exhibited an inverse trend to iron, showing a significant ($p \leq 0.05$) increase with the incorporation of Sacha inchi flour. In formulations without substitution, zinc levels were around 0.5 mg/100 g, whereas in those with complete substitution, zinc levels reached up to 2.6 mg/100 g, representing a 4.6-fold increase compared to formulations without Sacha inchi flour. This finding suggests that Sacha inchi flour is a substantial source of zinc, an essential mineral involved in metabolic functions and immune system regulation.

The results indicate that the replacement of wheat flour with Sacha inchi cake flour and sunflower oil with Sacha inchi oil has a significant impact on the mineral profile of the cookies. The incorporation of Sacha inchi flour enhances the content of essential macrominerals such as calcium, magnesium, potassium, and phosphorus, which may offer nutritional benefits, particularly for populations at risk of deficiencies in these nutrients. However, the observed reduction in iron content suggests that additional sources of this mineral may be required to prevent potential deficiencies in the final product. From a technological perspective, the stability of sodium content and the increase in zinc levels may be advantageous, as zinc plays a critical role in enzymatic functions and protein structural stability. Moreover, the high phosphorus content could improve certain functional properties of the dough, such as water retention capacity and structural integrity of the final product.

## Total polyphenol content and antioxidant capacity of cookies formulated with Sacha inchi flour and oil

The influence of different substitutions on the contribution of compounds with functional properties was analyzed. **Table 2** presents the results of the analysis of total polyphenol content and antioxidant activity in cookies formulated with different percentages of wheat flour and sunflower oil replacement using Sacha inchi cake flour and Sacha inchi oil. The results indicate that the phenolic compound content increased significantly ($p \leq 0.05$) with flour and oil substitution, reaching up to 1.46 times that of the control (formulation without substitution) in the formulation with 100% wheat flour replacement. Significant differences ($p \leq 0.05$) were observed between treatments with different levels of flour substitution; however, within the same flour level, oil substitution did not have a significant effect. These findings support previous reports indicating that Sacha inchi cake flour is a primary source of polyphenols and they are consistent with studies showing increased polyphenol content in other baked products, such as bread, where the incorporation of Sacha inchi flour also enhanced the levels of bioactive compounds [16] Furthermore, they align with the high content of these biocomponents reported in the bromatological analysis previously conducted on this sacha inchi cake flour [13]. This suggests that its use as a substitute for wheat flour boosts the contribution of these compounds, thereby improving the functional properties of the formulations, particularly due to its antioxidant potential and valuable nutritional profile.

From an antioxidant activity perspective, the different formulations showed promising results. According to FRAP assays, a progressive increase in antioxidant capacity of up to 1.99 times was observed as the substitution levels of both Sacha inchi flour and oil increased to 100%. Significant differences ($p \leq 0.05$) were found between formulations with different levels of flour replacement, as well as between those with varying degrees of oil substitution. This suggests that both defatted Sacha inchi cake flour and oil contributed antioxidant compounds with free radical-scavenging activity. While no

direct evidence was found of polyphenol contributions from the oil, primarily from the cake flour, the increased antioxidant activity, whether due to flour or oil substitution, suggests that, in addition to the flour's effect, the oil may provide other antioxidant compounds such as carotenoids, chlorophylls, and fat-soluble vitamins. These results are consistent with findings from previous studies on other baked goods, such as bread, where the incorporation of Sacha inchi cake flour also enhanced antioxidant capacity, confirming its potential in functional food formulations [16]. These findings underline the functional potential of both Sacha inchi flour and oil, suggesting that replacing sunflower oil with Sacha inchi oil could not only provide a healthier lipid profile, but also improve the oxidative stability and antioxidant capacity of the final product. On the other hand, the results of the DPPH assay showed that the antioxidant capacity was higher in formulations with partial oil replacement compared to the control and those with Sacha inchi flour replacement. However, in formulations where both flour and oil were completely replaced, a significant decrease in antioxidant activity was observed ($p \leq 0.05$).

Overall, these results demonstrate that the partial or total replacement of wheat flour and sunflower oil with defatted Sacha inchi cake flour and Sacha inchi oil significantly influences the antioxidant activity of the cookies. Furthermore, the observed increase in polyphenols and antioxidant capacity reinforces the potential for Sacha inchi derivatives as functional ingredients. These findings highlight the potential of Sacha inchi flour and oil in baked goods, offering an alternative with antioxidant benefits and a healthier lipid profile.

### Effect of incorporating Sacha inchi oil and cake flour on physical properties of cookies

As commonly acknowledged, the diameter, thickness, and spread ratio of cookies are essential indicators of dough behavior during baking and directly influence consumer perception [51]. According to the data presented in Table 3 and Fig 2, the partial or complete substitution of wheat flour with Sacha inchi oil cake flour, and sunflower oil with Sacha inchi oil, had a significant impact on these physical attributes Cookie diameters ranged from 3.34 cm (50% oil cake flour and 50% sacha inchi oil) to 5.85 cm (0% oil cake flour and 100% sacha inchi oil). The control formulation (0% substitution) had a diameter of 5.41 cm, which was significantly ($p \leq 0.05$) higher than most of the formulations with Sacha inchi ingredients. Only the formulation with 100% Sacha inchi oil and no flour substitution showed a greater diameter than the control ($p \leq 0.05$), indicating that full replacement of sunflower oil with Sacha inchi oil promotes cookie spread. This observation is consistent with previous studies suggesting that oils with lower melting points enhance dough expansion during baking. In contrast, formulations containing Sacha inchi oil cake flour showed significantly reduced diameters ($p \leq 0.05$). The lowest value was

**Table 3. Dimensions, color parameters and hardness of the cookie's formulations.**

| Formulation | Diameter (cm) | Thickness (cm) | Spread factor | L* | a* | b* | ΔE | Hardness (N) |
|---|---|---|---|---|---|---|---|---|
| Ctl | 5.41±0.02[c] | 1.88±0.02[b] | 2.88±0.02[e] | 57.92±1.09[de] | 1.91±0.06[c] | 28.93±0.25[bc] | -- | 51.74±6.16[bc] |
| F1 | 4.60±0.07[e] | 1.24±0.02[e] | 3.70±0.03[d] | 65.86±0.65[a] | 1.52±0.76[c] | 31.67±1.83[a] | 8.64±1.13[a] | 64.04±7.38[b] |
| F2 | 5.85±0.08[a] | 0.90±0.03[f] | 6.48±0.14[a] | 61.23±1.44[bc] | 1.58±0.15[c] | 29.30±0.54[bc] | 3.37±0.33[de] | 128.12±6.89[a] |
| F3 | 5.68±0.09[b] | 1.44±0.03[d] | 3.94±0.10[c] | 59.41±1.53[cde] | 2.96±0.37[b] | 30.47±0.16[ab] | 2.04±0.04[f] | 19.06±4.13[d] |
| F4 | 3.34±0.05[g] | 1.27±0.04[e] | 2.64±0.12[f] | 58.60±2.06[cde] | 0.80±0.02[d] | 23.52±0.77[d] | 6.11±0.66[b] | 19.95±6.90[d] |
| F5 | 5.03±0.06[d] | 0.91±0.05[f] | 5.49±0.22[b] | 56.76±3.47[e] | 5.33±0.43[a] | 31.19±2.10[a] | 5.44±1.17[bc] | 32.49±7.56 cd |
| F6 | 4.64±0.02[e] | 1.93±0.04[a] | 2.40±0.06[g] | 63.08±1.55[ab] | 3.03±0.24[b] | 28.49±0.23[c] | 4.48±0.34 cd | 22.01±3.32 cd |
| F7 | 3.58±0.04[f] | 1.25±0.02[e] | 2.87±0.05[e] | 60.08±0.55 cd | 2.12±0.57[c] | 27.83±0.63[c] | 3.19±0.81[ef] | 8.90±2.07[d] |
| F8 | 4.59±0.07[e] | 1.55±0.04[c] | 2.96±0.07[e] | 60.64±1.05[bcd] | 1.99±0.39[c] | 27.79±0.60[c] | 3.03±0.42[ef] | 8.94±2.70[d] |

Data represent the mean±standard deviation. Different superscript letters indicate significantly different mean values ($p<0.05$).

**Ctl:** 0% sacha inchi oil cake flour, 0% sacha inchi oil; **F1:** 0% oil cake flour, 50% sacha inchi oil; **F2:** 0% oil cake flour, 100% sacha inchi oil; **F3:** 50% oil cake flour, 0% sacha inchi oil; **F4:** 50% oil cake flour, 50% sacha inchi oil; **F5:** 50% oil cake flour, 100% sacha inchi oil; **F6:** 100% oil cake flour, 0% sacha inchi oil; **F7:** 100% oil cake flour, 50% sacha inchi oil; **F8:** 100% oil cake flour, 100% sacha inchi oil.

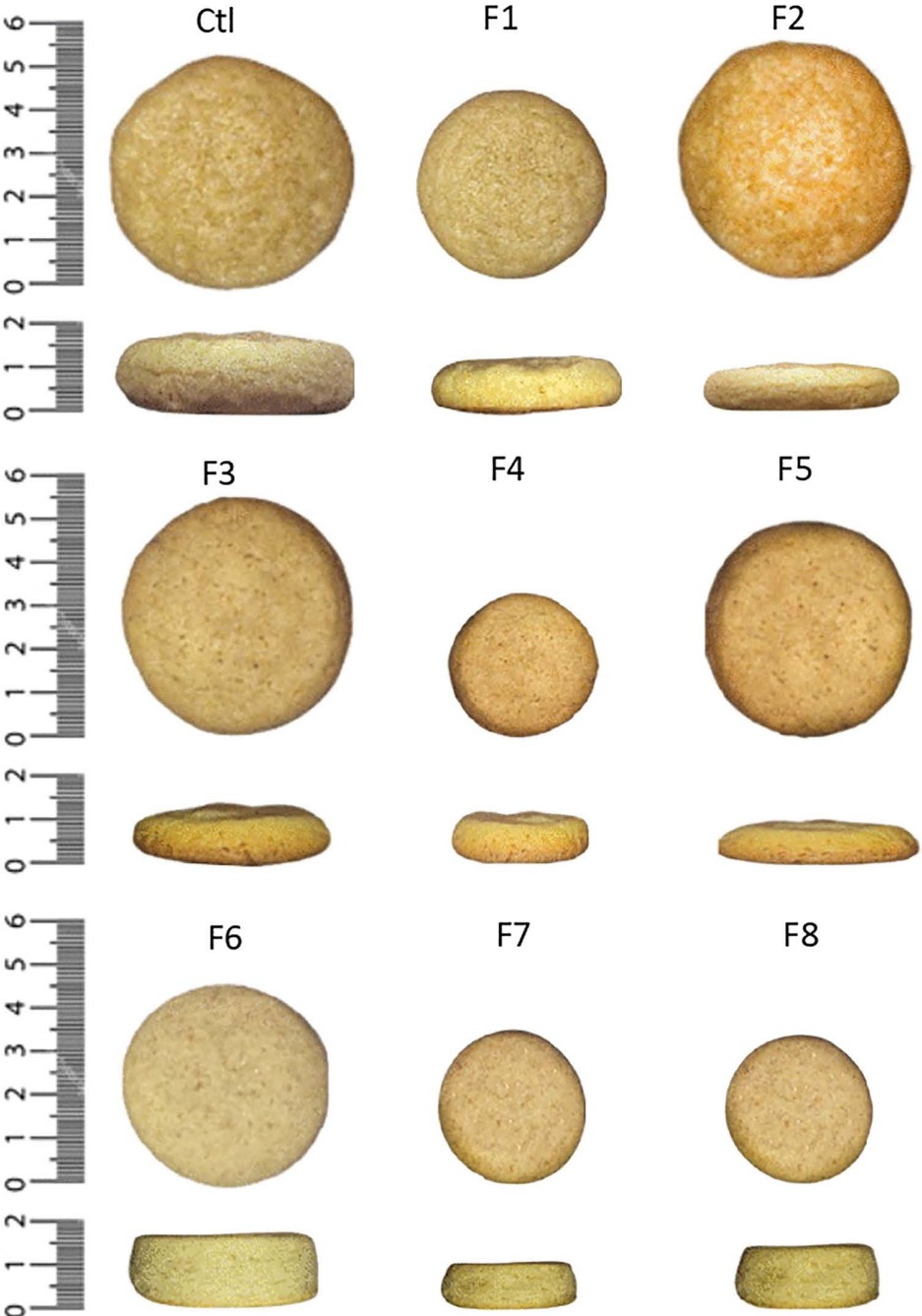

**Fig 2. Sides and tops appearance of cookies formulations.** Ctl: 0% sacha inchi oil cake flour, 0% sacha inchi oil; F1: 0% oil cake flour, 50% sacha inchi oil; F2: 0% oil cake flour, 100% sacha inchi oil; F3: 50% oil cake flour, 0% sacha inchi oil; F4: 50% oil cake flour, 50% sacha inchi oil; F5: 50% oil cake flour, 100% sacha inchi oil; F6: 100% oil cake flour, 0% sacha inchi oil; F7: 100% oil cake flour, 50% sacha inchi oil; F8: 100% oil cake flour, 100% sacha inchi oil.

observed in the formulation with 50% oil cake flour and 50% Sacha inchi oil (3.34 cm), followed by that with 100% oil cake flour and 50% Sacha inchi oil (3.58 cm) and 100% oil cake flour and 0% oil (4.64 cm). This reduction is likely due to the increased fiber and protein content, which may elevate dough viscosity and hinder spreading during baking [52,53]. The thickest cookie (1.93 cm) ($p \le 0.05$) was observed in the formulation with 100% oil cake flour and 0% Sacha inchi oil, while the thinnest (0.90 cm) resulted from the formulation with 0% oil cake flour and 100% Sacha inchi oil. Overall, oil-rich formulations without oil cake flour tended to produce thinner cookies, whereas those with higher oil cake flour content were thicker ($p \le 0.05$), likely due to restricted spread and increased dough structure. The spread ratio (diameter/thickness), an integrated indicator of expansion, ranged from 2.40 (100% oil cake flour and 0% sacha inchi oil) to 6.48 (0% oil cake flour and 100% sacha inchi oil). The control (0% substitution) had a moderate spread ratio of 2.88. Notably, the formulation with 0% oil cake flour and 100% Sacha inchi oil displayed the highest spread ratio ($p \le 0.05$), confirming sacha inchi oil's role in promoting expansion. On the other hand, combinations with 50% or 100% oil cake flour and little or no sacha inchi oil had the lowest spread ratios ($p \le 0.05$), consistent with their higher thickness and smaller diameters. This behavior may result from the fiber and protein in the oil cake flour reducing dough extensibility and interfering with leavening agents [54]. These results confirm a clear interaction between fat and flour type in shaping cookie geometry.

Cookie color is a key sensory attribute that influences consumer acceptance. Using the CIE Lab* system, lightness ($L*$), redness ($a*$), and yellowness ($b*$) were evaluated. Lightness values ranged from 56.76 (50% oil cake flour and 100% Sacha inchi oil) to 65.86 (0% oil cake flour and 50% Sacha inchi oil). Most formulations with Sacha inchi oil or oil cake flour showed higher $L*$ than the control (57.92) ($p \le 0.05$), indicating a lighter appearance, potentially due to natural pigments and reduced Maillard browning [38]. However, full flour substitution (100% oil cake flour) did not always yield the highest lightness, suggesting complex interactions during baking. The $a*$ parameter (red-green axis) ranged from 0.80 (50% oil cake flour and 50% Sacha inchi oil) to 5.33 (50% oil cake flour and 100% Sacha inchi oil). Redness generally increased with higher sacha inchi oil cake flour content, possibly due to enhanced Maillard reactions from additional amino acids. Formulations containing only Sacha inchi oil had less influence on $a*$, highlighting the greater impact of flour substitution ($p \le 0.05$). Regarding $b*$ (yellow-blue axis), values varied from 23.52 (50% oil cake flour and 50% sacha inchi oil) to 31.67 (0% oil cake flour and 50% Sacha inchi oil). Incorporation of sacha inchi oil slightly increased $b*$, while full flour substitution showed a minor decreasing effect. The formulation with 50% substitution of both oil and flour showed a high $b*$ value (31.19), suggesting a synergistic effect between flour and oil in enhancing yellowness. Nonetheless, some differences were not statistically significant, indicating that yellowness changes were formulation-dependent. The total color difference (ΔE) compared to the control confirmed perceptible variation across all samples. The highest ΔE was found in the formulation with 0% oil cake flour and 50% Sacha inchi oil (8.64) ($p \le 0.05$), followed by the combination with 50% oil cake flour and 50% Sacha inchi oil (6.11), while the substitution of only 50% oil cake flour (and 0% sacha inchi oil) showed the lowest deviation (2.04). These differences likely stem from both ingredient color and chemical reactions (Maillard and caramelization) modulated by the presence of protein and fat [55].

In terms of texture, cookie hardness varied significantly. The control cookie had a hardness of 51.74 N. Substitution with 50% Sacha inchi oil (no flour replacement) increased hardness to 64.04 N, while full replacement of oil (100% Sacha inchi oil, no flour substitution) resulted in the highest hardness (128.12 N), more than 2.4 times that of the control ($p \le 0.05$). This suggests that Sacha inchi oil leads to a denser structure, likely due to differences in fatty acid profiles and their effect on dough structure and moisture retention [56,57]. In contrast, formulations containing Sacha inchi oil cake flour showed a marked decrease in hardness ($p \le 0.05$). Substituting 50% of the wheat flour (with no Sacha inchi oil) led to a reduction to 19.06 N, while combinations with 100% flour substitution (with or without oil) resulted in values between 8.90 N and 22.01 N, indicating a much softer texture—up to 5.8 times less firm than the control ($p \le 0.05$). These reductions suggest that the absence of gluten and the lipid content in Sacha inchi flour contribute to a more tender structure. The combination of Sacha inchi oil and flour may also exert an emulsifying effect, softening the matrix even further, as reported in studies involving emulsifiers [58].

 

The results demonstrate that the type and proportion of fat and flour significantly affect cookie geometry, color, and texture. Sacha inchi oil increases spread and hardness, while Sacha inchi oil cake flour reduces both spread and hardness, enabling targeted modulation of cookie characteristics. From a technological perspective, these ingredients offer potential for developing customized cookies with desirable texture and visual appeal, although adjustments may be required in applications where firmer textures are needed.

## Conclusions

This study demonstrated that replacing wheat flour and sunflower oil with Sacha inchi derivatives significantly enhances cookies' nutritional profile and functional properties. Nutritionally, higher substitution levels increased protein, dietary fiber, omega-3 fatty acids, and essential minerals (calcium, magnesium, potassium, phosphorus, and zinc) while reducing carbohydrates and saturated fats. However, the observed decrease in iron content indicates the need for complementary iron sources in final formulations. From a functional perspective, incorporating Sacha inchi oil cake flour increased polyphenol content and antioxidant capacity, supporting its potential as an ingredient for cardiovascular and metabolic health benefits. Physical assessment revealed that Sacha inchi oil increased cookie diameter and spread, while the oil cake flour reduced diameter and increased thickness, resulting in softer cookies with full flour replacement. These changes provide opportunities for tailoring baked goods with specific sensory attributes. Overall, these findings position Sacha inchi flour and oil as promising ingredients for developing functional, nutritious, and sustainable bakery products while promoting the valorization of agro-industrial by-products and Amazonian crops in alignment with circular economy principles. Future research should address texture modifications at high substitution levels, evaluate consumer acceptance of novel flavors, establish reliable supply chains beyond small-scale farming, assess economic viability compared to conventional ingredients, and navigate regulatory requirements for novel food ingredients. Addressing these challenges through targeted research will be essential to fully realize the commercial potential of these valuable Amazonian by-products.

## Acknowledgments

We would like to thank Universidad San Francisco de Quito USFQ, particularly the USFQ Research Dean's Office, for supporting Fernando E. Alejandro Ruiz and Julio F. Ortega Jácome, students of Food Engineering. We also sincerely thank the members of the Shushufindi community in Sucumbíos Province, located in the northern Amazon region of Ecuador, for their valuable collaboration in providing samples of sacha inchi seeds, oil cake, and oil. Their support and generosity were essential to the development of this study.

## Author contributions

**Conceptualization:** Paola Vásconez Duchicela, Julio Vásconez Espinoza, Pablo Beltrán-Ayala, María J. Andrade-Cuvi, José M. Alvarez-Suarez.

**Data curation:** José R. Mora, Andrea C. Landázuri, María J. Andrade-Cuvi, José M. Alvarez-Suarez.

**Formal analysis:** Fernando E. Alejandro Ruiz, Julio F. Ortega Jácome, Margarita G. Chancay Pinargote, María J. Andrade-Cuvi.

**Funding acquisition:** Andrea C. Landázuri, Paola Vásconez Duchicela, Julio Vásconez Espinoza, José M. Alvarez-Suarez.

**Investigation:** José R. Mora, Andrea C. Landázuri, María J. Andrade-Cuvi, José M. Alvarez-Suarez.

**Methodology:** José R. Mora, María J. Andrade-Cuvi, José M. Alvarez-Suarez.

**Project administration:** Pablo Beltrán-Ayala, José M. Alvarez-Suarez.

**Resources:** José R. Mora, Paola Vásconez Duchicela, Julio Vásconez Espinoza, Pablo Beltrán-Ayala, José M. Alvarez-Suarez.

**Supervision:** José R. Mora, María J. Andrade-Cuvi, José M. Alvarez-Suarez.

**Validation:** José R. Mora, María J. Andrade-Cuvi, José M. Alvarez-Suarez.

**Visualization:** María J. Andrade-Cuvi, José M. Alvarez-Suarez.

**Writing – original draft:** Fernando E. Alejandro Ruiz, Julio F. Ortega Jácome, Margarita G. Chancay Pinargote, Paola Vásconez Duchicela, Julio Vásconez Espinoza.

**Writing – review & editing:** José R. Mora, Andrea C. Landázuri, Pablo Beltrán-Ayala, María J. Andrade-Cuvi, José M. Alvarez-Suarez.

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
