## [Decision Letter · Decision Letter 0]

6 Oct 2025

Dear Dr. Alvarez-Suarez,

Thank you for submitting your manuscript to PLOS ONE. After careful consideration, we feel that it has merit but does not fully meet PLOS ONE’s publication criteria as it currently stands. Therefore, we invite you to submit a revised version of the manuscript that addresses the points raised during the review process.

We look forward to receiving your revised manuscript.

Kind regards,

Academic Editor

PLOS ONE

Journal Requirements:

“This work was supported by Universidad San Francisco de Quito through an Interdisciplinary Grant (Project ID: 23218) awarded to José M. Alvarez-Suárez.”

4. We note that your Data Availability Statement is currently as follows: All relevant data are within the manuscript and in Supporting Information files.

Reviewers' comments:

Reviewer's Responses to Questions

**Comments to the Author**

1. Is the manuscript technically sound, and do the data support the conclusions?

Reviewer #1: Partly

Reviewer #2: Yes

Reviewer #3: Partly

2. Has the statistical analysis been performed appropriately and rigorously?

Reviewer #1: Yes

Reviewer #2: Yes

Reviewer #3: No

3. Have the authors made all data underlying the findings in their manuscript fully available?

Reviewer #1: Yes

Reviewer #2: Yes

Reviewer #3: Yes

4. Is the manuscript presented in an intelligible fashion and written in standard English?

Reviewer #1: Yes

Reviewer #2: Yes

Reviewer #3: Yes

Reviewer #1: Based on a critical review of the manuscript titled "Influence of partial or total substitution of wheat flour and sunflower oil with Sacha inchi (Plukenetia volubilis L.) flour and oil on the quality and nutritional properties of cookies", here are detailed comments section-wise, including line and page numbers

Line 39: The phrase “reaching 19.65%, more than double the control” is impactful but would benefit from a statistical reference (e.g., p-value).

Line 43: The decrease in iron content by 25% is mentioned, but the nutritional implications of this reduction are not discussed.

Line 45: “According with the lipid profiling” should be corrected to “according to the lipid profiling.”

Line 49: The contrast between Sacha inchi flour reducing hardness and oil increasing it is interesting—consider briefly explaining the mechanism.

Line 53: The phrase “suggesting interactions between lipid, protein, and fiber components” is vague—consider specifying the nature of these interactions.

Line 56: The phrase “valorization of underutilized Amazonian crops” is strong—consider briefly mentioning sustainability or economic impact.

Line 71: The phrase “monoculture-based grain production” could benefit from a citation or example.

Line 93: “Particularly recognized for its remarkable protein and lipid profile” is a strong claim—consider referencing specific studies.

Line 98: The term “underutilized by-product” is appropriate, but the potential safety concerns of using oil cake in food should be acknowledged.

Line 106: “Inconsistent extraction efficiency” is mentioned—quantitative data or references would strengthen this point.

Line 111: “Significant gap in understanding” is a strong claim—consider citing a recent review or meta-analysis to support it.

Line 119: The phrase “cardioprotective benefits” should be supported by a reference or clarified as a hypothesis.

Line 149 (Page 14): Table 1 is referenced but not discussed in detail—consider summarizing key differences in formulations.

Line 175 (Page 15): The energy calculation formula is correct, but the assumption of 4 kcal/g for protein and carbs and 9 kcal/g for fat should be explicitly stated.

Line 203 (Page 16): The GC-MS quantification method assumes a 1:1 response factor—this should be justified or validated.

Line 215 (Page 17): Recovery rates (84.70%–99.96%) are excellent—consider stating the acceptable range for validation.

Line 229 (Page 18): The use of gallic acid equivalents is standard, but the calibration curve R² value should be reported.

Line 269 (Page 19): The use of Tukey’s test is appropriate—ensure all comparisons in tables reflect this (some values lack superscripts).

Reviewer #2: The study is quite well-prepared and detailed. However, a few points caught my attention. In the recipe for the cookie formulation, were the ingredients used on a dry basis? Secondly, why was no sensory evaluation performed? Including sensory evaluation could have been valuable, as it would allow the produced cookies to be assessed in terms of important criteria such as texture and consumer acceptability.

Reviewer #3: Structure the abstract into four distinct sections. For example, include a reference: https://doi.org/10.1371/journal.pone.0307506.

Incorporate a flow diagram illustrating the cookie formulation process. For example, include a reference: https://doi.org/10.1371/journal.pone.0307506.

Organize Table 1 following the format in the provided reference for clarity and ease of understanding: https://doi.org/10.1016/j.foohum.2025.100771.

Include the table and accompanying text in the main manuscript, as they are critical components. For example, adhere to the referenced format.

Specify the equipment details (model, brand, country) in the Materials and Methods section.

Provide the AOAC method number for each analytical determination.

Report the absorbance values used in spectrophotometric measurements, along with the quantification equation and the coefficient of determination.

Indicate the concentrations of solutions prepared in this section.

Conduct a sensory analysis, as it is vital for this study.

Include references in the methodology section for procedures such as color determination and hardness measurements.

In Table 2, report either moisture content or water activity, as including both is unnecessary.

**Do you want your identity to be public for this peer review?** For information about this choice, including consent withdrawal, please see our Privacy Policy

Reviewer #1: No

Reviewer #2: **Yes: ** Nazlı Şahin

Reviewer #3: No

---

## [Author Response · Author response to Decision Letter 1]

16 Oct 2025

Journal Requirements:

RESPONSE: We confirm that the manuscript has been revised to fully comply with PLOS ONE’s style and formatting requirements. All elements—including structure, section headings, file naming conventions, and manuscript layout—have been carefully adapted according to the official templates and author guidelines provided on the journal’s website. We have ensured that the revised submission adheres to the formatting standards indicated in the links supplied by the Editorial Office.

RESPONSE: We thank the Editorial Office for the clarification. Following your instructions, we have carefully reviewed the entire manuscript and removed all funding-related information from the text. The funding details now appear exclusively in the Funding Statement section of the online submission form, as requested.

“This work was supported by Universidad San Francisco de Quito through an Interdisciplinary Grant (Project ID: 23218) awarded to José M. Alvarez-Suárez.”

RESPONSE: The statement "The funders had no role in study design, data collection and analysis, decision to publish, or preparation of the manuscript." was included in the cover letter

4. We note that your Data Availability Statement is currently as follows: All relevant data are within the manuscript and in Supporting Information files.

RESPONSE: We confirm that the submission provides all the statistically processed data required to replicate the results presented in the study. Although raw, unprocessed instrumental outputs are not included, all values used for the statistical analyses—such as means, standard deviations, and the numerical data underlying the figures and tables

RESPONSE: We appreciate the comment. However, this point is no longer applicable. The data originally included as supplementary information were published separately. Therefore, in the revised version, we have removed these supplementary files and now refer directly to the published article where these results are available. Consequently, the titles of these files are no longer relevant, and the in-text citations have been updated to reflect the formal publication of these data.

Reviewer #1: Based on a critical review of the manuscript titled "Influence of partial or total substitution of wheat flour and sunflower oil with Sacha inchi (Plukenetia volubilis L.) flour and oil on the quality and nutritional properties of cookies", here are detailed comments section-wise, including line and page numbers

Line 39: The phrase “reaching 19.65%, more than double the control” is impactful but would benefit from a statistical reference (e.g., p-value).

Line 43: The decrease in iron content by 25% is mentioned, but the nutritional implications of this reduction are not discussed.

Line 45: “According with the lipid profiling” should be corrected to “according to the lipid profiling.”

Line 49: The contrast between Sacha inchi flour reducing hardness and oil increasing it is interesting—consider briefly explaining the mechanism.

Line 53: The phrase “suggesting interactions between lipid, protein, and fiber components” is vague—consider specifying the nature of these interactions.

Line 56: The phrase “valorization of underutilized Amazonian crops” is strong—consider briefly mentioning sustainability or economic impact.

RESPONSE: We thank the reviewer for their thoughtful comments. In response to Reviewer 3’s recommendations, the Abstract has been revised to comply with the PLOS ONE format (Background, Methodology, Results, and Conclusions). As a result of this restructuring, the specific phrases referred to by the reviewer have been removed and are therefore no longer present in the revised version of the Abstract. Furthermore, since the Abstract must follow a defined structure and a limited word count, the additional details suggested by the reviewer (e.g., statistical references, mechanistic explanations, and nutritional or sustainability implications) will be appropriately addressed in the Results and Discussion section of the manuscript, where they can be presented with sufficient depth and context.

Line 71: The phrase “monoculture-based grain production” could benefit from a citation or example.

RESPONSE: We thank the reviewer for this helpful observation. Following this suggestion, an example and a supporting reference have been added to clarify and strengthen the statement. The revised sentence now reads as follows:

The baking industry, which has traditionally relied on refined wheat flour and conventional vegetable oils, now faces multiple challenges, including the prevalence of gluten intolerance, rising cases of metabolic disorders [2], and the environmental impact of monoculture-based grain production, such as intensive wheat and maize cultivation, which contributes to soil degradation, biodiversity loss, and increased greenhouse gas emissions [3].

- FAO. The State of the World’s Biodiversity for Food and Agriculture. Rome; 2020.

Line 93: “Particularly recognized for its remarkable protein and lipid profile” is a strong claim—consider referencing specific studies.

RESPONSE: We thank the reviewer for this valuable observation. To substantiate this statement, we have included two supporting references that highlight both the growing scientific attention and commercial relevance of Sacha inchi due to its nutritional composition, bioactive potential, and applications in food and health industries. The revised sentence is now properly supported by the following references:

- Wang S, Zhu F, Kakuda Y. Sacha inchi (Plukenetia volubilis L.): Nutritional composition, biological activity, and uses. Food Chemistry. 2018;265: 316–328. doi:10.1016/j.foodchem.2018.05.055

- Kodahl N. Sacha inchi (Plukenetia volubilis L.)—from lost crop of the Incas to part of the solution to global challenges? Planta. 2020;251: 80. doi:10.1007/s00425-020-03377-3

Line 98: The term “underutilized by-product” is appropriate, but the potential safety concerns of using oil cake in food should be acknowledged.

RESPONSE: We thank the reviewer for this important observation. Following this valuable suggestion, we have included a clarification in the text to acknowledge the potential safety concerns related to the presence of antinutritional compounds in the raw oil cake and to highlight that these can be effectively reduced through appropriate processing. The revised text now reads as follows:

Nevertheless, despite its promising nutritional composition, the raw oil cake may contain certain antinutritional compounds, including alkaloids, oxalates, nitrates, tannins, thiocyanates, saponins, phytic acid, trypsin inhibitors, and glucosinolates, which can affect protein digestibility and mineral bioavailability if not properly processed. These compounds are, however, largely reduced through technofunctional processing steps such as roasting, drying, extrusion, or baking, as well as hydrothermal treatments (autoclaving or hot air), which have been shown to significantly decrease their levels in Sacha inchi oil cake [11].

- Landines Vera E, Villacrés E, Coello Ojeda K, Guadalupe Moyano V, Quezada Tobar M, Quelal MB, et al. Evaluation of antinutrients, nutritional, and functional properties in sacha inchi (Plukenetia volubilis L) cake treated with hydrothermal processes. Heliyon. 2024;10: e37291. doi:10.1016/j.heliyon.2024.e37291

Line 106: “Inconsistent extraction efficiency” is mentioned—quantitative data or references would strengthen this point.

RESPONSE: We thank the reviewer for this helpful comment. Following their suggestion, a supporting reference has been added to provide quantitative context and substantiate the statement regarding extraction efficiency variability. The revised sentence now reads as follows:

Despite the promising nutritional profile of Sacha inchi, several practical challenges limit the commercial utilization of its by-products, including variable extraction efficiencies ranging from 70% to 90% depending on seed origin and processing conditions [15], underdeveloped supply chains, and potential cost disadvantages compared to conventional ingredients.

- Vásquez-Ocmín PG, Freitas Alvarado L, Sotero Solís V, Paván Torres R, Mancini-Filho J. Chemical characterization and oxidative stability of the oils from three morphotypes of Mauritia flexuosa L.f, from the Peruvian Amazon. Grasas y Aceites. 2010;61: 390–397. doi:10.3989/gya.010110.

Line 111: “Significant gap in understanding” is a strong claim—consider citing a recent review or meta-analysis to support it.

RESPONSE: We thank the reviewer for this constructive comment. To substantiate this statement, we have added a recent review by Goyal et al. (2022), which provides comprehensive evidence that current Sacha inchi research is largely focused on oil composition and extraction parameters, while limited attention has been given to studies evaluating the simultaneous use of Sacha inchi oil and flour in food formulations. The revised sentence now reads as follows:

While interest in Sacha inchi as a functional ingredient is growing, research examining the simultaneous application of both its flour and oil as direct replacements for traditional baking ingredients remains limited, creating a significant gap in understanding their combined effects in everyday food products. Recent reviews have highlighted that most studies on Sacha inchi focus primarily on the compositional and physicochemical properties of the oil, with comparatively scarce investigations addressing the integration of both fractions—oil and flour—within food matrices or processing systems [16]. This lack of integrated research constrains a comprehensive understanding of the functional and technological synergies between lipid and protein components derived from this underutilized Amazonian crop.

Line 119: The phrase “cardioprotective benefits” should be supported by a reference or clarified as a hypothesis.

RESPONSE: We thank the reviewer for this accurate observation. In response, the sentence has been revised to avoid any possible misunderstanding, as the study did not directly evaluate cardioprotective effects. Instead, our objective was to develop formulations with a health-beneficial fatty acid profile, characteristic of Sacha inchi oil. Since this section of the manuscript outlines the study’s objectives, we consider it inappropriate to include bibliographic references at this point. Therefore, a general descriptive term has been used, which clearly conveys the intended meaning without the need for citation. The revised sentence now reads as follows:

Additionally, we analyzed fatty acid profiles to obtain formulations with a health-beneficial lipid composition, characteristic of Sacha inchi oil, and measured polyphenol content and antioxidant capacity to quantify the functional properties these Amazonian ingredients contribute to the final products.

Line 149 (Page 14): Table 1 is referenced but not discussed in detail—consider summarizing key differences in formulations.

RESPONSE: We thank the reviewer for this helpful observation. In response, a brief summary of the main formulation differences has been added to the text to provide clearer context for Table 1. The revised paragraph now highlights the three substitution gradients applied—(i) partial and total replacement of sunflower oil with Sacha inchi oil, (ii) 50 % replacement of wheat flour with oilcake flour combined with variable oil levels, and (iii) complete replacement of wheat flour with oilcake flour combined with increasing Sacha inchi oil content. This addition clarifies the experimental design and allows readers to better understand the progressive incorporation of Sacha inchi ingredients across formulations. The revised sentence now reads as follows:

As summarized in Table 1, the formulations were designed to progressively increase the proportion of Sacha inchi ingredients, enabling the evaluation of both single and combined substitution effects. This resulted in three main substitution gradients: (i) replacement of sunflower oil with Sacha inchi oil (F1–F2); (ii) simultaneous 50 % substitution of wheat flour with oilcake flour and variable levels of oil (F3–F5); and (iii) total substitution of wheat flour with oilcake flour and increasing Sacha inchi oil content (F6–F8). The constant ingredients (egg, sugar, honey, salt, vanilla, and sodium bicarbonate) ensured that observed variations could be attributed primarily to differences in oil and flour composition.

Line 175 (Page 15): The energy calculation formula is correct, but the assumption of 4 kcal/g for protein and carbs and 9 kcal/g for fat should be explicitly stated.

RESPONSE: We thank the reviewer for this precise observation. In response, the text has been revised to explicitly indicate the conversion factors used in the energy value calculation. The following clarification has been added below Equation (2):

Conversion factors of 9 kcal/g for fat and 4 kcal/g for protein and carbohydrates were applied according to the Atwater system [26].

- Maclean WC, Harnly JM, Chen J, Chevassus-Agnes S, Gilani G, Livesey G, et al. Food energy—

---

## [Decision Letter · Decision Letter 1]

26 Nov 2025

Dear Dr. Alvarez-Suarez,

Thank you for submitting your manuscript to PLOS ONE. After careful consideration, we feel that it has merit but does not fully meet PLOS ONE’s publication criteria as it currently stands. Therefore, we invite you to submit a revised version of the manuscript that addresses the points raised during the review process.

We look forward to receiving your revised manuscript.

Kind regards,

Karthikeyan Venkatachalam, Ph.D.

Academic Editor

PLOS ONE

Journal Requirements:

Reviewers' comments:

Reviewer's Responses to Questions

**Comments to the Author**

Reviewer #2: All comments have been addressed

Reviewer #3: All comments have been addressed

Reviewer #4: (No Response)

2. Is the manuscript technically sound, and do the data support the conclusions?

Reviewer #2: Yes

Reviewer #3: Yes

Reviewer #4: Yes

3. Has the statistical analysis been performed appropriately and rigorously?

Reviewer #2: Yes

Reviewer #3: Yes

Reviewer #4: Yes

4. Have the authors made all data underlying the findings in their manuscript fully available?

Reviewer #2: Yes

Reviewer #3: Yes

Reviewer #4: Yes

5. Is the manuscript presented in an intelligible fashion and written in standard English?

Reviewer #2: (No Response)

Reviewer #3: Yes

Reviewer #4: Yes

Reviewer #2: No additional comment

Reviewer #3: This manuscript is well-written, with clear and concise prose that effectively communicates its key ideas. All reviewer comments have been addressed thoroughly, demonstrating a strong understanding of the feedback provided. Overall, it meets the standards for publication and contributes meaningfully to the field.

Reviewer #4: Summary – Manuscript PONE-D-25-18191R1

The present work investigates Sacha inchi (Plukenetia volubilis L.), a plant native to the Amazon basin, for its role as a primary ingredient in the process of baking cookies for the purpose of sustainable, health conscious, large scale food production. The work suggests that Sacha inchi, which is a rare seed oil, increases the macronutrient composition of cookies in favor of a more healthy nutritional profile. Given rising concerns with prevalence of metabolic disorder, dysbiosis arising from overconsumption of highly-processed foods, and an expanding agricultural/environmental dilemma from current practices, this is a timely and important study looking to shift the paradigm of industrial processes with a basis in indigenous nutritional practices immemorial. This work is supported by a body of literature that has previously described the nutrient composition of Sacha inchi, which is rich in omega-3 fatty acids and its refined by-products are similarly rich in nutritive chemicals; the authors astutely point to some of the undesired by-products and the current processes of removal and refinement that render these to be relatively safe. Sacha inchi and its by-products were used as replacements for wheat flour and sunflower oil, then products were generated by modulating the amounts of Sacha inchi products in the formulation - materials and products were evaluated for nutrient content, lipid composition, polyphenol and antioxidant activities, and physical properties. The study is promising for shifting the diversity of nutritional sources in large-scale food processing and leverages existing ingredients with presence in the historical lexicon. The study is limited by its evaluation of only the products at end-stage and using a single baking method, which may require refinement specific to the individual compositions. The expected benefits of each of Sacha inchi’s components are well-described, but the combination of these nutritional factors, their chemical interactions, and their potential bioavailability remain unknown. As well, the physical parameters suggest a subjective score representing desirability of the final product, however the roles for taste and texture in the eating experience are not described, as this would have included a human subjects component. This supports future studies looking into subjective experience using optimized formulations with human cohorts, and does not reduce enthusiasm for this report. Please see my detailed comments.

Major Comments:

Evaluation of lipid composition is a strength of this work, however further dissection of omega-3 and omega-6 lipid species (e.g., docosahexaenoic acid, eicosapentaenoic acid, arachidonic acid, etc.) and short chain fatty acids would further strengthen this work.

Subjective response to the various cookie formulations would provide additional rationale for the physical outcomes measured. As well, features such as taste and texture (beyond rigidity) are not described, and will play a major role with incorporating Sacha inchi into regular use.

Minor Comments:

The results section of the abstract indicates fold change for large effects and percentage values for smaller effects. This may be made more consistent by using the percentage notation throughout but is acceptable. “Six-fold” higher should be rewritten with the numerals, “6-fold”, to match the subsequent effects sizes (i.e., 10.8-fold, 10.2-fold, etc.).

The claim in Lines 101-103, “…traditionally consumed by indigenous communities for their health benefits and nutrient density.”, require additional citation. Are there accounts of the particular ingredients, described herein, being used medicinally?

**Do you want your identity to be public for this peer review?** For information about this choice, including consent withdrawal, please see our Privacy Policy

Reviewer #2: **Yes: ** Nazlı Şahin

Reviewer #3: **Yes: ** Md. Asaduzzaman

Reviewer #4: **Yes: ** Donovan A Argueta

---

## [Author Response · Author response to Decision Letter 2]

16 Dec 2025

Reviewer 1

In view of this manuscript, the data given about Sacha inch is not sufficient. Give some detail about this composition, etc.

Response: We appreciate the reviewer’s observation and agree that additional compositional detail strengthens the contextualization of Sacha inchi within the introduction. Accordingly, we have expanded this section by incorporating a concise summary of its key nutritional and chemical attributes. The revised text now reads as follows:

Sacha inchi is recognized for its exceptional chemical composition, with seeds containing 33–54% oil, 24–30% protein, and an optimal ω-6/ω-3 ratio close to 1:1, driven by high levels of α-linolenic acid (∼47%) and linoleic acid (∼38%) [13]. Recent characterization of Ecuadorian materials confirms this fatty acid profile and additionally shows that the defatted oilcake is highly concentrated in protein (~42%), dietary fiber, and essential minerals, including calcium (2,63 mg/kg), magnesium (3,69 mg/kg), potassium (7,68 mg/kg), and phosphorus (9,14 mg/kg), while retaining meaningful levels of polyphenols and antioxidant activity [14]. These compositional attributes, together with favorable techno-functional properties such as water absorption and swelling capacity, support the use of both Sacha inchi oil and oilcake flour as nutrient-dense, sustainable ingredients for bakery formulations.

Some common mistakes like incorrect spellings, grammatical mistakes are also present in this manuscript. Kindly correct all these.

Response: We thank the reviewer for pointing this out. The entire manuscript has now been thoroughly revised for language quality, grammar, spelling, and academic style. A professional native-English proofreader has reviewed and corrected the full text to ensure clarity, consistency, and linguistic accuracy.

In GC MS Analysis, there is no citation reference is given in overall paragraph.

Response: We appreciate the reviewer’s observation. We would like to clarify that the methodology is indeed properly cited in the paragraph immediately preceding the GC–MS description. Specifically, the manuscript states:

“Fatty acid content analysis was performed following the internal standard method described by Ponce et al. and based on AC Analytical Controls Application Note 1410 [32,33].”

These references cover the analytical procedure used and provide the methodological foundation for the GC–MS analysis presented. Nonetheless, we have reviewed this section to ensure that the citation placement is clear and unambiguous.

Again, there is no citation is given in color determination method, hardness determination and statistical analysis. Give proper citation in overall manuscript.

Response: We thank the reviewer for this observation and sincerely apologize for the unintentional omission of the corresponding citations in the sections describing the color determination and hardness determination methods. In the revised manuscript, we have now added the appropriate methodological references to ensure proper attribution and methodological accuracy.

Regarding the statistical analysis, we would like to clarify that the manuscript already includes a dedicated subsection titled “Design of Experiment and Statistical Analysis” within the Materials and Methods section, where all statistical procedures, software, and analytical criteria are fully described. We have verified that this section clearly communicates the statistical framework applied in our study.

In addition, we have thoroughly reviewed the entire manuscript to ensure that all methodological sections contain their complete and correct references, and to confirm that no methodological citation is missing in the revised version.

Please write all references according to Journal guidelines in Vancouver style, mistakes like non abbreviated journals names, space issues etc are present, deeply attention is required in this regard.

Response: We thank the reviewer for highlighting this issue. All references in the manuscript have now been thoroughly revised and corrected to fully comply with the journal’s Vancouver style guidelines, including proper journal name abbreviations, spacing, punctuation, and overall formatting. We have carefully checked the entire reference list and all in-text citations to ensure complete consistency with the required style.

Reviewer 4

Summary – Manuscript PONE-D-25-18191R1

The present work investigates Sacha inchi (Plukenetia volubilis L.), a plant native to the Amazon basin, for its role as a primary ingredient in the process of baking cookies for the purpose of sustainable, health conscious, large scale food production. The work suggests that Sacha inchi, which is a rare seed oil, increases the macronutrient composition of cookies in favor of a more healthy nutritional profile. Given rising concerns with prevalence of metabolic disorder, dysbiosis arising from overconsumption of highly-processed foods, and an expanding agricultural/environmental dilemma from current practices, this is a timely and important study looking to shift the paradigm of industrial processes with a basis in indigenous nutritional practices immemorial. This work is supported by a body of literature that has previously described the nutrient composition of Sacha inchi, which is rich in omega-3 fatty acids and its refined by-products are similarly rich in nutritive chemicals; the authors astutely point to some of the undesired by-products and the current processes of removal and refinement that render these to be relatively safe. Sacha inchi and its by-products were used as replacements for wheat flour and sunflower oil, then products were generated by modulating the amounts of Sacha inchi products in the formulation - materials and products were evaluated for nutrient content, lipid composition, polyphenol and antioxidant activities, and physical properties. The study is promising for shifting the diversity of nutritional sources in large-scale food processing and leverages existing ingredients with presence in the historical lexicon. The study is limited by its evaluation of only the products at end-stage and using a single baking method, which may require refinement specific to the individual compositions. The expected benefits of each of Sacha inchi’s components are well-described, but the combination of these nutritional factors, their chemical interactions, and their potential bioavailability remain unknown. As well, the physical parameters suggest a subjective score representing desirability of the final product, however the roles for taste and texture in the eating experience are not described, as this would have included a human subjects component. This supports future studies looking into subjective experience using optimized formulations with human cohorts, and does not reduce enthusiasm for this report. Please see my detailed comments.

Major Comments:

- Evaluation of lipid composition is a strength of this work, however further dissection of omega-3 and omega-6 lipid species (e.g., docosahexaenoic acid, eicosapentaenoic acid, arachidonic acid, etc.) and short chain fatty acids would further strengthen this work.

Response: We appreciate the reviewer’s suggestion regarding the need to further clarify the presence or absence of specific omega-3 and omega-6 lipid species, as well as short-chain fatty acids. In response, we have incorporated the following paragraph into the Fatty acid profile and content of cookies formulated with Sacha inchi flour and oil section:

“No long-chain omega-3 fatty acids (such as EPA and DHA) or long-chain omega-6 fatty acids (e.g., arachidonic acid) were detected in any of the formulations. This is consistent with the previously reported fatty acid profile of Sacha inchi oil, which is predominantly α-linolenic acid (C18:3, ω-3) together with linoleic acid (C18:2, ω-6), and lacks long-chain omega-3 PUFA such as EPA and DHA. Similarly, no short-chain fatty acids (C4:0–C10:0) were observed in our chromatograms, which is also consistent with previous reports for oils from this seed [8,17].”

This addition clarifies the absence of these fatty acids in our formulations and aligns our findings with previously reported compositional data for Sacha inchi oil.

- Subjective response to the various cookie formulations would provide additional rationale for the physical outcomes measured. As well, features such as taste and texture (beyond rigidity) are not described, and will play a major role with incorporating Sacha inchi into regular use.

Response: We appreciate the reviewer’s observation regarding the value of incorporating sensory or subjective evaluations to complement the physical and compositional results. We fully agree that sensory perception—including taste, aroma, mouthfeel, and overall acceptability—plays a crucial role in determining the practical relevance and consumer integration of new formulations containing Sacha inchi.

However, conducting any form of sensory evaluation, even at an exploratory level, requires prior approval from an accredited institutional ethics committee, as human participation is involved. Unfortunately, ethical approval for sensory testing was not included in the scope of the original study design. In accordance with PLOS ONE’s policies, the inclusion of human sensory data is strictly contingent upon documented approval by an institutional review board or ethics committee. Because this approval was not obtained beforehand, we are unable to incorporate sensory evaluation results in the current manuscript.

We would also like to note that this point was raised by previous reviewers, who fully understood our explanation regarding the ethical limitations and the impossibility of adding sensory data at this stage. We respectfully hope that the current reviewer will similarly appreciate the rationale for the absence of sensory analyses in this study.

Minor Comments:

- The results section of the abstract indicates fold change for large effects and percentage values for smaller effects. This may be made more consistent by using the percentage notation throughout but is acceptable. “Six-fold” higher should be rewritten with the numerals, “6-fold”, to match the subsequent effects sizes (i.e., 10.8-fold, 10.2-fold, etc.).

Response: We thank the reviewer for this helpful observation regarding consistency in numerical notation within the abstract. In accordance with the suggestion, we have revised the expression “six-fold higher” to “6-fold higher” to ensure uniformity with the subsequent effect sizes reported (e.g., 10.8-fold, 10.2-fold, 3-fold). We have also harmonized fold-change terminology throughout the abstract to improve clarity and consistency. The revised abstract now reflects a coherent numerical format.

“Cookies containing Sacha inchi flour showed significant increases in protein (up to 19.65%), fat, fiber (6-fold higher), ash, and energy, with a reduction in carbohydrates. Mineral content increased markedly for calcium (10.8-fold), magnesium (10.2-fold), potassium (3-fold), phosphorus (5.4-fold), and zinc (4.6-fold), while iron decreased by 25%. Lipid profiling revealed a higher proportion of polyunsaturated fatty acids, particularly α-linolenic acid (omega-3), and lower saturated fats in cookies containing Sacha inchi oil. The incorporation of Sacha inchi ingredients also increased total polyphenols (up to 1.46-fold) and antioxidant activity (up to 1.99-fold). Texture analysis showed that Sacha inchi flour reduced hardness (up to 5.8-fold softer than the control), whereas Sacha inchi oil increased firmness (up to 2.4-fold). Full replacement of sunflower oil increased cookie diameter and spread ratio, while Sacha inchi flour reduced diameter and increased thickness. Color parameters were also affected, reflecting compositional and Maillard-related changes during baking.”

- The claim in Lines 101-103, “…traditionally consumed by indigenous communities for their health benefits and nutrient density.”, require additional citation. Are there accounts of the particular ingredients, described herein, being used medicinally?

Response: We thank the reviewer for this important clarification request. In response, we have added an appropriate citation documenting the traditional consumption and reported health-related uses of Sacha inchi among Indigenous communities. This reference has now been incorporated into the revised manuscript to support the statement regarding its traditional relevance and perceived benefits.

Rios M, Koziol MJ, Borgtoft Pedersen H, Granda G, editors. Plantas Útiles del Ecuador: Aplicaciones, Retos y Perspectivas / Useful Plants of Ecuador: Applications, Challenges, and Perspectives. Quito: Ediciones Abya-Yala; 2007.

Cárdenas Sierra DM, Gómez Rave LJ, Soto JA. Biological Activity of Sacha Inchi (Plukenetia volubilis Linneo) and Potential Uses in Human Health: A Review. Food Technol Biotechnol. 2021;59: 253–266. doi:10.17113/ftb.59.03.21.6683

---

## [Editor Report · Decision Letter 2]

17 Dec 2025

Influence of partial or total substitution of wheat flour and sunflower oil with Sacha inchi (Plukenetia volubilis L.) flour and oil on the quality and nutritional properties of cookies

PONE-D-25-18191R2

Dear Dr. Alvarez-Suarez,

We’re pleased to inform you that your manuscript has been judged scientifically suitable for publication and will be formally accepted for publication once it meets all outstanding technical requirements.

Kind regards,

Karthikeyan Venkatachalam, Ph.D.

Academic Editor

PLOS One
---

## [Editor Report · Acceptance letter]

PONE-D-25-18191R2

PLOS One

Dear Dr. Alvarez-Suarez,

I'm pleased to inform you that your manuscript has been deemed suitable for publication in PLOS One. Congratulations! Your manuscript is now being handed over to our production team.

Kind regards,

on behalf of

Dr. Karthikeyan Venkatachalam

Academic Editor

PLOS One